# Sea surface temperature in the Indian sector of the Southern Ocean over the Late Glacial and Holocene

Lisa C. Orme[1,2], Xavier Crosta[3], Arto Miettinen[1,4], Dmitry V. Divine[1], Katrine Husum[1], Elisabeth Isaksson[1], Lukas Wacker[5], Rahul Mohan[6], Olivier Ther[3], Minoru Ikehara[7]

[1]Norwegian Polar Institute, Tromsø, 9296, Norway
[2]ICARUS, Department of Geography, Maynooth University, Maynooth, Ireland
[3]UMR 5805 EPOC, Université de Bordeaux, Bordeaux, France
[4]Ecosystems and Environment Research Programme, University of Helsinki, Helsinki, Finland
[5]Department of Physics, ETH Zürich, 8093 Zürich, Switzerland
[6]National Centre for Polar and Ocean Research (NCPOR), Ministry of Earth Sciences, Vasco-da-Gama, Goa 403804, India
[7]Center for Advanced Marine Core Research, Kochi University, Nankoku, 783-8502, Japan

*Correspondence to*: Lisa C. Orme (lisa.orme@mu.ie)

**Abstract.** Centennial and millennial scale variability of Southern Ocean temperature over the Holocene is poorly known, due to both short instrumental records and sparsely distributed high-resolution temperature reconstructions, with evidence for past temperature variations in the region coming mainly from ice core records. Here we present a high-resolution (~60 year), diatom-based sea-surface temperature (SST) reconstruction from the western Indian sector of the Southern Ocean that spans the interval 14.2 to 1.0 ka BP (calibrated kiloyears before present). During the late deglaciation, the new SST record shows cool temperatures at 14.2-12.9 ka BP and gradual warming between 12.9-11.6 ka BP in phase with atmospheric temperature evolution. This supports that the evolution of the Southern Ocean SST during the deglaciation was linked with a complex combination of processes and drivers associated with reorganisations of atmospheric and oceanic circulation patterns. Specifically, we suggest that Southern Ocean surface warming coincided, within the dating uncertainties, with the reconstructed slowdown of the Atlantic Meridional Overturning Circulation (AMOC), rising atmospheric $CO_2$ levels, changes in the southern westerly winds and enhanced upwelling. During the Holocene the record shows warm and stable temperatures from 11.6-8.7 ka BP followed by a slight cooling and greater variability from 8.7 to 1 ka BP, with a quasi-periodic variability of 200-260 years identified by spectral analysis. We suggest that the increased variability during the mid- to late Holocene reflects the establishment of centennial variability in SST connected with changes in the high latitude atmospheric circulation and Southern Ocean convection.

## 1 Introduction

Research into the sequence of events associated with the deglacial period has highlighted that there were contrasting patterns of millennial-scale warming and cooling between the southern and northern high latitudes, considered to have resulted from global-scale processes, namely fluctuations in the strength of the AMOC and latitudinal shifts in the major atmospheric

circulation cells. For example, during the Younger Dryas (12.9-11.7 ka BP; Rasmussen et al., 2014) a prolonged weakening of the AMOC reduced the northward heat transport in the Atlantic allowing heat to accumulate in the Southern Ocean causing a "bipolar seesaw" characterised by opposite ocean temperature anomalies between the two hemispheres (Denton et al., 2010 and references therein). These temperature changes caused a rapid alteration of the atmospheric circulation, with a southward shift of the Intertropical Convergence Zone (ITCZ) and Southern Westerly Winds (SWW). A combination of processes including enhanced eddy transport of heat across the Antarctic Circumpolar Current (ACC), upwelling of warm water and dissipation of sea ice then resulted in warming of the Southern Ocean and Antarctica (Levermann et al., 2007; Screen et al., 2009; Denton et al., 2010). Finally, the increased upwelling in the Southern Ocean enhanced $CO_2$ outgassing and thus warming globally. Despite many of these processes occurring within the Southern Ocean the strongest evidence for the patterns and magnitude of southern high latitude warming during the Younger Dryas comes from Antarctic ice cores (e.g. Stenni et al., 2011). Although a number of records from the Southern Ocean show warming during the Younger Dryas, the timing, magnitude and duration of warming are far from consistent between records (Bianchi and Gersonde, 2004; Divine et al., 2010; Siani et al., 2013; Xiao et al., 2016). In the same vein, the cooler Antarctic Cold Reversal (ACR, 14.7-12.7 ka BP; Stenni et al., 2011), preceding the Younger Dryas, similarly displays a strong regional heterogeneity that is not fully understood (Pedro et al., 2015). Acquiring high-resolution proxy records of past sea surface temperature (SST) is therefore important for establishing the spatio-temporal patterns of warming in the Southern Ocean and their link with Antarctic temperature variability.

During the Holocene the majority of records from the open ocean of the Atlantic and Indian sectors of the Southern Ocean show that the early Holocene, defined as the period 11.7-8.2 ka BP (Walker et al., 2018), was warmer than the subsequent mid to late Holocene (e.g. Bianchi and Gersonde, 2004; Nielsen et al., 2004; Anderson et al., 2009; Divine et al., 2010; Xiao et al., 2016). The suggested causes of the temperature evolution during the Holocene include variations in southern hemisphere insolation (e.g. Shevenell et al., 2011; Xiao et al., 2016), in the interplay of southern and northern hemisphere insolation (e.g. Nielsen et al., 2004) and in the weakening of the AMOC in response to northern hemisphere deglaciation (Renssen et al., 2010).

There is also evidence of centennial variability of southern high latitude climate through the Holocene as shown by some palaeoclimate records, including those from the Southern Ocean (e.g. Leventer et al., 1996; Crosta et al., 2007; Katsuki et al., 2012) and those reflecting atmospheric circulation changes such as the Southern Annular Mode (SAM) (e.g. Moreno et al., 2014; Turney et al., 2016). Observational and modelling evidence have suggested that centennial fluctuations in deep convection in the Southern Ocean (Weddell Sea) lead to variability in the surface water temperature of the Southern Ocean and the strength of the atmospheric circulation (Latif et al., 2013). Others have linked the oscillations to AMOC variability (Debret et al., 2009) or solar activity (e.g. Leventer et al., 1996; Crosta et al., 2007; Turney et al., 2016). To understand further how the Southern Ocean temperature has varied during the Holocene, there is a need for high-resolution records capturing centennial to multi-centennial scale variations.

We here present a new diatom-based SST reconstruction from the western Indian sector of the Southern Ocean spanning the last 14.2 thousand years with an average temporal resolution of 60 years, to show the patterns, timing and magnitude of temperature variability through the late deglaciation and the Holocene.

## 2 Study Area

        The Conrad Rise is located in the west Indian sector of the Southern Ocean (around 54 °S, 40°E, Figure 1) close to
the Atlantic Sector. The rise is <3000 m depth, with two sea mounts (Ob Bank and Lena Seamount) at depths of 300 m (Ansorge et al., 2008). The Conrad Rise is situated within the diatom ooze belt, which circles Antarctica, extending from the Polar Front in the north to the late winter/spring sea ice extent in the south (Diekmann, 2007).

        Today the site is located within the Permanent Open Ocean Zone (POOZ), with the mean winter sea ice limit lying to the south at 59°S (Park et al., 1998; Parkinson and Cavalieri, 2012). The mean summer SST at 10 m depth is 3.5°C (Locarnini
et al., 2013), while summer measurements show that at the Conrad Rise the surface mixed layer extends to 70-80 m depth with temperatures of 2-2.5 °C and salinity of 33.90 – 33.95 psu (Park et al., 1998; Anilkumar et al., 2006). The surface oceanographic characteristics are likely influenced by the atmospheric climate and circulation. The core of the SWW has been situated north of the Conrad Rise at approximately 52 °S during the period 1979-2010 AD (Swart and Fyfe, 2012). Across the Southern Ocean positive SAM anomalies cause stronger, southward shifted westerlies, and this also alters SST, causing cooling
in the Antarctic Zone due to stronger Ekman transport of cold Antarctic surface water to the north (Lovenduski and Gruber, 2005).

        The Conrad Rise is situated within the Antarctic Circumpolar Current (ACC) and close to the Polar Front, which is defined by the maximum northern extent of the 2°C isotherm between 100-300 m. Some modelling and observations support that two branches of the Polar Front flow to the north and south of the Conrad Rise, transporting 35 and 25 Sv of water,
respectively, with low flow velocities over the rise itself (Figure 1b; Pollard and Read, 2001; Ansorge et al., 2008; Sokolov and Rintoul, 2009). However, assessments of the Polar Front position based on identification of a subsurface temperature minima indicate that it is in fact situated to the north, passing between the Conrad Rise and Crozet Islands at approximately 50°S (Pauthenet et al., 2018). Topography has a strong influence on the position and form of the ACC in this region. At the Conrad Rise, branches of the Polar Front become topographically trapped and therefore have low spatial variability through
time (Pollard and Read, 2001; Pauthenet et al., 2018).

## 3 Materials and methods

### 3.1 Core sampling and sediment description

The 2.48 m long core KH-10-7 COR1GC was retrieved from the Conrad Rise (54.2673°S, 39.7663°E, Fig. 1, water depth of 2834 m) using a gravity corer during the expedition KH-10-7 on board the R/V *Hakuho-Maru* in 2010.

The sediment consisted of homogenous diatom ooze; from 2.48 to 2.2 m the sediment colour was greenish-grey, while above this the sediment had a pale yellow colour. At approximately 5 cm depth there was a dark grey horizon.

### 3.2 Age model

Fifteen AMS radiocarbon dates were obtained on mono-specific samples of the planktic foraminifera *Neogloboquadrina pachyderma* sinistral. The core chronology was constructed in Bacon version 2.3.3 (Blaauw and Christen, 2011), an age-depth modelling technique that uses Bayesian statistics to reconstruct accumulation histories for deposits, by combining radiocarbon and other dates with prior information. This analysis used the Marine13 calibration dataset (Reimer et al., 2013). The shape of sedimentation rate and memory strength parameters used in the Bacon modelling were set to 1.5 and 4, respectively.

To correct the raw radiocarbon ages we used a constant reservoir age of $890 \pm 100$ years recommended for this region (http://radiocarbon.LDEO.columbia.edu/) (Butzin et al., 2005). However, we acknowledge the possibility that the reservoir age may have varied through time, particularly during the deglaciation when ventilation of deep waters occurred. In the Atlantic sector of the Southern Ocean, the comparison of radiocarbon ages and $^{226}$Ra decay in barite suggests a change in regional reservoir ages between the early and mid to late Holocene of ~800 years (van Beek et al., 2002). In the Pacific sector, Holocene reservoir ages have been estimated in the Ross Sea through a comparison of $^{14}$C and U-Th dating of corals, which showed limited changes in reservoir age during the last 6 ka BP (Hall et al., 2010). In the east Pacific sector (core MD07-3088, Figure 1), a comparison between radiocarbon and tephra dating of marine sediments suggests that at 14 ka BP and 11.5 ka BP the reservoir age was 975-920 years, while in the mid-late Holocene it was 800 years (Siani et al., 2013), indicating a change of just 100-175 years. While this study was located north of the Subantarctic Front in a different oceanographic setting to the Conrad Rise, the MD07-3088 study site is bathed in waters originating from the Southern Ocean that are deflected north along the coast of South America, supporting that changes here reflect Southern Ocean reservoir age variability. Given the information delivered by these remote studies and the lack of evidence from the Indian sector of the Southern Ocean, we believe it is more sensible to apply a constant reservoir age correction, but acknowledge that there is greater age uncertainty in the late deglacial section of the COR1GC reconstruction.

**3.3 Diatom analysis**

Diatom analysis was conducted at ~1 cm intervals. Slide preparation and counting of over 300 diatom valves per sample followed standard methods (Crosta and Koç, 2007). Summer SST (at 10 m depth) and winter sea-ice concentration (WSIC) were estimated using the Modern Analogue Technique (MAT) applied to diatom assemblages (Crosta et al., 2020). This method uses a modern database composed of 249 surface sediment samples. Modern summer (January-February-March) SSTs from 10 m depth were interpolated on a 1° grid from the World Ocean Atlas 2013 (Locarnini et al., 2013) using Ocean

Data View (Schlitzer, 2014). Winter sea ice concentration (WSIC), represented by the month of September when sea ice is at its maximum extent around Antarctica (Parkinson, 2019), was interpolated on a 1° grid from the numerical atlas of Schweitzer (1995). The MAT was implemented from the "bioindic" package (Guiot and de Vernal, 2011) built on the R-platform (http://cran.r-project.org/) and used the relative abundances of 32 diatom species and the chord distance, as a similarity metric, to select the five most similar modern analogs. The threshold above which modern analogs are supposed to be too dissimilar

to the fossil assemblage is fixed at the first quartile of random distances on the validation/modern dataset. Quantitative estimates of summer SST and WSIC are a similarity-weighted mean of the SST or WSIC associated with the selected modern analogs (Guiot et al., 1993). This method yields a $R^2$ of 0.96 and a root mean square error of prediction (RMSEP) of ~1°C for summer SST and a $R^2$ of 0.92 and a RMSEP of ~10% for WSIC.

        The relative abundance of different diatom species and predictions of SST and WSIC may be altered by differential

dissolution and preservation between species. While temperature is the dominant control on diatom species assemblages (Pichon et al., 1992; Esper et al., 2010), the dissolution of weakly silicified species, which are often those from cold waters, could potentially result in warmer reconstructed temperatures or an underestimation of sea-ice extent (Xiao et al., 2016). However, as the dissolution of species is strongest in the Subantarctic Zone (SAZ) and sea-ice zone of the Southern Ocean (Esper et al., 2010) and a dissolution minima (of 10%) is centred on 55°S (Pichon et al., 1992), the Conrad Rise is a region

likely to be less influenced by diatom dissolution than locations at both lower and higher latitudes in the Southern Ocean. Furthermore, several studies have shown that the surface water gradients in most abundant diatom species are preserved in the surface sediments (Gersonde and Zielinski, 2000; Armand et al., 2008a,b; Cárdenas et al., 2019). The strong association between ocean surface conditions and diatom assemblages within the sediment is supported by several statistical techniques that have been used to quantitatively infer past surface conditions using diatom analysis (e.g. Gersonde and Zielinski, 2000;

Crosta et al., 2004; Esper and Gersonde, 2014).

        The total diatom flux (DF), or diatom accumulation rate, was calculated to provide an approximation of productivity through time, following the method described in Romero et al. (2015). To calculate the DF, the number of valves per gram of dry sediment (diatom absolute abundance) was multiplied by the bulk density of wet sediment and the sedimentation rate. The resulting value was then divided by two to represent the number of diatoms rather than valves. The sedimentation rates and

diatom abundances were calculated for core COR1GC, however because density was not measured on core COR1GC the wet bulk density measurements for core COR1bPC were instead used (Oiwane et al., 2014), as this core was recovered from the

same location. The density measurements were transferred from COR1bPC to COR1GC using the chronologies of the two cores.

### 3.4 SiZer analysis

SiZer (Significance of Zero Crossings of the Derivative) (Chaudhuri and Marron, 1999) is a scale-space technique that was applied to explore statistically significant features in the reconstructed SST. A key idea in SiZer is that significant features are found at different time scales, that is, at different levels of data smoothing. This makes it particularly useful in paleoclimate studies since the salient features in a timeseries may depend heavily on the time horizon on which it is analysed. This method has been used previously in a number of palaeoclimate studies (e.g. Divine et al., 2010).

In SiZer the observed data are viewed at varying levels of resolution while the notion of scale is controlled through the bandwidth h in the local linear kernel estimator. For each scale h and time t of the signal, a test is performed to see whether the smoothed data series has a local derivative significantly different from zero, in other words, to see if the slope at a specific time point for a given scale is significantly different from zero. The Gaussian kernel estimator embedded in SiZer does not require an analysed timeseries to be evenly spaced, the method is therefore applied to the data directly without any prior 165 resampling. SiZer visualises the output of the analysis in a feature map where the results are displayed as a function of time and scale.

### 3.5 Spectral analysis

     Many geophysical time series have distinctive red noise characteristics that can be modelled by a first order autoregressive (AR1) process. REDFIT spectral analysis (Schulz and Mudelsee, 2002) was carried out on the Holocene section 170 of the SST reconstruction to establish whether quasi-periodic variability existed. This method of spectral analysis was selected as it has been designed for unevenly spaced paleoclimate data, therefore avoiding the need to resample and interpolate the data, which can introduce bias in the inferred spectrum (Schulz and Mudelsee, 2002). The analysis was conducted using the REDFIT tool integrated in the PAST 3.25 software (Hammer et al., 2001) using the Welch window with two overlapping segments (Schulz and Mudelsee, 2002). The appropriateness of the AR1 model to describe the analysed data was tested using 175 a nonparametric runs test (Bendat and Piersol, 1986) embedded in the package.

### 4 Results

     According to the constructed chronology the core covers the period from 14.2-1 ka BP and has a mean sedimentation rate of ~20 cm ka$^{-1}$ (Table 1; Figure 2). The sedimentation rate decreased from 14 to 12 ka BP when it stabilised at its lowest values until 10 ka BP. It subsequently increased over the course of the Holocene to reach values >20 cm ka$^{-1}$ during the late 180 Holocene.

The diatom assemblage of COR1GC is dominated by pelagic open ocean taxa, particularly *Fragilariopsis kerguelensis* and *Thalassiosira lentiginosa* (Zielinski and Gersonde, 1997; Crosta et al., 2005), with accompanying species typical of the Polar Front Zone (PFZ) and POOZ (Figure 3). Species from the PFZ (*F. kerguelensis*, *Thalassionema nitzschioides* var *lanceolata*) and Subantarctic Zone (SAZ) (*Shionodiscus oestrupii*, previously known as *Thalassiosira oestrupii*; Crosta et al., 2005; Romero et al., 2005), were more abundant in the interval 12-9.5 ka BP, with lowest relative abundances at 14.2-12 ka BP and 4-1 ka BP. Conversely, POOZ species abundances (especially *Thalassiothrix antarctica*, *Thalassiosira gracilis*, *Fragilariopsis separanda* and *Fragilariopsis rhombica*; Zielinski and Gersonde, 1997; Crosta et al., 2005) were lower at 14.2-9.5 ka BP, after which point they gradually increased in proportion over the Holocene. Diatom species associated with sea ice, including *Fragilariopsis ritscheri*, *Fragilariopsis curta*, *Fragilariopsis cylindrus*, *Porosira glacialis*, *Fragilariopsis obliquecostata* and *Thalassiosira tumida* (Zielinski and Gersonde, 1997; Armand et al., 2005), had individually low average abundances of ≤0.3%. These species have been grouped together (Figure 3). The sea ice species were present in low abundances through the record, but were slightly more abundant between 14.2-12.6, 7.6-6.2 and 3.9-1 ka BP and lower from 12.6-7.6 and 6.2-3.9 ka BP.

The diatom flux record of COR1GC (Figure 3) shows a range from ~2000 to 14,000 million diatoms cm$^2$ ka$^{-1}$. From 14.2 to 12 ka BP the diatom flux decreased (from ~12,000 to 2000 million diatoms cm$^2$ ka$^{-1}$). Concentrations remained low between 12 and 10.3 ka BP followed by higher fluxes with an average of 7200 million diatoms cm$^2$ ka$^{-1}$ between 10.3 and 5.2 ka BP. After 5.2 ka BP fluxes decreased, with an average of 4500 million diatoms cm$^2$ ka$^{-1}$ between 5.2 and 1 ka BP.

The SST reconstruction for COR1GC shows that during the 14.2 - 12.9 ka BP period the average temperature was 2.9 °C with a range of 1.8-4.1 °C and standard deviation of 0.6 °C (Figure 4), which was followed by a phase of warming between 12.9 and 11.6 ka BP when temperatures increased from 2.1 to 5 °C. Between 11.6 and 8.7 ka BP SST was on average 4.3 °C, with a range of 3.3-5 °C and a standard deviation of 0.4 °C. After 8.7 ka BP the SST record shows a slightly lower average of 3.9 °C, a range of 2.2-5.2 °C, a standard deviation of 0.7 °C and a slight cooling trend. The record shows that during the slightly warmer interval at 11.6-8.7 ka BP the temperature varied less than after 8.7 ka BP, indicating a potentially more stable climate during the early Holocene and a more variable climate during the mid to late Holocene. SiZer analysis flags three events as statistically significant: the two short-term cooling events at ~8.2 ka BP and ~2.2 ka BP and the warming trend during the deglaciation, the latter of which is marked as significant when identifying variability over a broad range of timescales (Figure 5). Conversely, SiZer did not find the cooling through the Holocene to be significant. Spectral analysis conducted using the REDFIT method shows that during the Holocene there were quasi-periodic oscillations of 220 and 260 years, significant above the 95% false alarm level (Figure 6).

Summer sea ice did not reach the Conrad Rise through the last 14.2 ka BP (reconstruction not shown) and winter sea ice concentrations were very low and below the RMSEP throughout the period (Figure 4). These data however suggest slightly more extensive sea ice cover between 14 and 12 ka BP and after 8 ka BP, and absent winter sea ice between 12.2 and 8 ka BP when SSTs were ~1°C higher than today. Relatively low values of the inferred winter sea ice concentration suggest only

sporadic and relatively short-term (perhaps decadal to multi-decadal scale) expansions of winter sea ice to the coring location throughout the mid-late Holocene.

## 5 Discussion

### 5.1 Southern high latitude temperature

The millennial trends in the COR1GC temperature record are similar to those of SST records from the Atlantic sector of the Southern Ocean (Nielsen et al., 2004; Anderson et al., 2009; Divine et al., 2010; Xiao et al. 2016; Figure 7 A and B) within the age uncertainty of each record. Many of these show cooler temperatures at ~14.2-12.9 ka BP during the ACR (14.7-12.7 ka BP; Stenni et al., 2011), gradual warming during the Younger Dryas, high temperatures between 11.6 and 8.7 ka BP during the early Holocene, followed by a cooling trend thereafter. Another SST reconstruction from the Conrad Rise (PS2606-6; Figure 7C; Xiao et al., 2016), with lower temporal resolution than the COR1GC record, shows warm temperatures during the ACR and the early Holocene along with a slight cooling during the Younger Dryas, at odds with the other records from the POOZ. We suggest that the opposite trend in the PS2606-6 record during the late deglaciation could be related to either chronological issues, as it is the only SST record that is out-of-phase during this period, or the result of different reconstruction techniques. Furthermore, while most records show cooling over the Holocene, either gradually or as an abrupt cooling at the end of the early Holocene (Xiao et al., 2016), there are some differences between records. For example, the cooling in the COR1GC is small compared to other records as the change in mean temperature from the early to mid-late Holocene was ~0.4°C, although superimposed on this long-term trend were centennial events with 1-2°C cooler temperatures during the mid-late Holocene (Figure 4). We note, on the other hand, that core TN057-17 close to the Polar Front in the Atlantic sector shows a warming trend since ~4 ka BP (Figure 7; Nielsen et al., 2004), in contrast to a nearby low resolution record (cores PS1654/OPD1093) which indicates that SSTs were relatively stable over the late Holocene (Xiao et al., 2016). Despite many records having similar long-term patterns, these results therefore suggest some regional heterogeneity of temperature evolution at the centennial to multi-millennial scales through the late deglacial and Holocene.

The records from the Bouvet Island region, in the eastern Atlantic sector of the Southern Ocean, and the Conrad Rise (Figure 7 A-D; cores TN057-17-PC1, TN057-13-PC4 and PS2606-6; Xiao et al., 2016) show that the magnitude of warming between the ACR and early Holocene was 1-2°C. Nevertheless, there are variations in the exact magnitudes of the temperature fluctuations between the different records. Some previous records suggested that the SST during the ACR and the mid-late Holocene were similar (Nielsen et al., 2004; Anderson et al., 2009; Divine et al., 2010; Xiao et al., 2016; Figure 7 A-C). Our new record from core COR1GC however shows SSTs were 1°C lower during the ACR compared to the mid-late Holocene, in agreement with reconstructions of air temperature from ice cores (Figure 7E-F). The contrasting findings between SST reconstructions may be explained by the reconstructed temperatures being close to the prediction error of SST transfer functions, which is ~1°C in this study and 0.86°C in Xiao et al. (2016).

The millennial temperature evolution of the COR1GC record during the deglacial and early Holocene closely resembles the patterns of atmospheric temperature and sea ice extent, reconstructed from Antarctic ice core water isotope records and sodium flux records respectively (Figure 7 E and F; EPICA community members, 2006; Fischer et al., 2007; Jouzel et al., 2007; Stenni et al 2011). This is particularly evident when comparing the smoothed and normalised reconstructions of temperature ($\delta^{18}$O) and regional sea ice extent (ssNa) from the closest ice core, EDML, with the COR1GC SST record (Supplementary Information 1). The start of the COR1GC record at 14.2 ka BP is after the beginning of the ACR (at 14.7 ka BP; Stenni et al., 2011) meaning that the timing of the onset of the ACR at the Conrad Rise cannot be established. However despite some degree of dating uncertainty the warming from 12.9 - 11.6 ka BP appears closely aligned with the timing of warming identified in the ice core records at 12.7-11.9 ka BP (Stenni et al., 2011). A good synchronicity between Southern Ocean SST and Antarctic temperature evolution over the deglacial was also shown by a planktic foraminifera-based SST reconstruction from core MD07-3088 from the eastern Pacific sector, which like the COR1GC record showed cold temperatures during the ACR, followed by gradual warming between c.12.7 and 11.5 ka BP leading to slightly warmer conditions through the Holocene (Siani et al., 2013; Haddam et al., 2018). The MD07-3088 record was constrained by both radiocarbon dates and tephra horizons minimising dating uncertainties (Siani et al., 2013), which supports the chronological framework of core COR1GC and the new SST record. The marginally warmer early Holocene temperatures that followed at 12-8.7 ka BP are observed widely in ice core, terrestrial and coastal records from across Antarctica (Stenni et al., 2011; Verleyen et al., 2011; Figure 7E and 7F). Together this evidence suggests that the sea surface and atmospheric temperatures across the southern high latitudes varied synchronously during the deglacial to Holocene transition.

## 5.2 Causes of temperature variability

The SST variations may have been due to a number of interrelated factors, including in response to atmospheric temperature changes, variations in the AMOC, changes in the strength and position of the SWW and shifts in the location of the ACC fronts relative to the core site. Here these causes will be discussed in relation to the Younger Dryas and Holocene climate patterns, with the suggested causes summarised in Table 2.

### 5.2.1 Younger Dryas

Southern high latitude warming during the Younger Dryas is widely considered to be the result of a slowdown in the AMOC (McManus et al., 2004; Figure 8A), which caused reduced northward heat transport and the accumulation of heat in the southern hemisphere (Broecker, 1998; Stocker and Johnsen, 2003). It is considered that this hemispheric temperature change caused a southward shift of the ITCZ (Figure 8B; Hughen et al., 1996), which in turn caused poleward-shifted and strengthened SWW (Denton et al., 2010; Pedro et al., 2018). Evidence from the Southern Ocean supports that the SWW were stronger over a longer interval from 14 to 11 ka BP but only shifted southwards to a more poleward position at 12.5 ka BP (Fig 8C; Fletcher and Moreno, 2011; Saunders et al., 2018).

The changes to the SWW during the Younger Dryas are thought to have warmed the Southern Ocean in a number of ways, including through the southward migration of the Polar Front, causing local warming in regions currently to the south of the Polar Front (e.g. Barker et al., 2009) and through strengthening of the eddy transport of heat across the ACC (e.g. Pedro et al., 2018) causing a more widespread warming of the POOZ. Although there is a lack of palaeo-proxy evidence for stronger eddy transport of heat across the ACC, this mechanism has been identified in both model simulations and modern observations (Meredith and Hogg, 2006; Screen et al., 2009). It is considered that this initial warming of the Southern Ocean resulted in reduced sea ice in the Indian and Atlantic sectors at 12 ka BP (Bianchi and Gersonde, 2004; Iizuka et al., 2008), promoting further ocean warming in the newly ice-free region (Denton et al., 2010; Xiao et al., 2016; Pedro et al., 2018) and warming of the lower troposphere and Antarctica by ice-albedo feedbacks (Pedro et al., 2018). Greater upwelling, caused by stronger winds and a reduction in sea ice extent, has been shown by higher opal deposition to the south of the Polar Front in the Southern Ocean (Atlantic, Indian and Pacific sectors) through the period 12.7-11.5 ka BP (Figure 8D; Dezileau et al., 2003; Anderson et al., 2009; Siani et al., 2013). The combination of greater upwelling and reduced sea ice allowed more outgassing of $CO_2$ to the atmosphere (Figure 8F), causing warming globally (Monnin et al., 2001; Anderson et al., 2009; Clark et al., 2012; Saunders et al., 2018).

The warming and reduced sea ice shown by the COR1GC reconstructions during the Younger Dryas (Figure 3 and Figure 4) coincide with many of these changes, including the AMOC weakening, strengthened and southward-shifted SWW, increased upwelling, reduced sea ice, rising $CO_2$ and warming over Antarctica (Figure 8; Supplementary Information 1). However, there is no evidence of increased productivity and therefore upwelling at the Conrad Rise, as diatom fluxes instead decreased through this period (Figure 8E). Although the marine records have sources of age uncertainty, the similarity in the timing of changes in marine and terrestrial records supports that warming was a result of processes connected with the interhemispheric oceanic and atmospheric reorganisation (Figure 8).

One challenge is to establish the relative importance of each of the outlined causes of warming, because although changes occurred synchronously they may not have all contributed equally to warming of the Southern Ocean. For example, the Conrad Rise is located close to the modern Polar Front and therefore a southward shift of the SWW during the Younger Dryas may have pushed the Polar Front to the south, leading to warming. Indeed, a southward shift is indicated by an increase in the Polar Front species *Thalassionema nitzschioides* var. *lanceolata* at c.12 ka BP (Figure 3). Although observations and models show that the fronts of the Polar Front are topographically constrained in this region (Pollard and Read, 2001; Graham et al., 2012; Pauthenet et al., 2018), it is possible that larger magnitude changes in the SWW were able to divert the Polar Front to the south of the Conrad Rise. The relative importance of $CO_2$ in contributing to the warming of Antarctica and the Southern Ocean, which resulted from the above described changes in circulation and Southern Ocean conditions, is also not clear. Principle component analysis on a synthesis of global temperature reconstructions spanning the last deglaciation indicate that temperatures globally were primarily controlled by greenhouse gas concentrations (Clark et al., 2012) and temperature over Antarctica rose synchronously with $CO_2$ increases during the Younger Dryas (Parrenin et al., 2013). The synchronicity between

the $CO_2$ record and some of the temperature records (COR1GC, MD07-3088 and ice core records) over the deglaciation, suggests that the $CO_2$ increase, following the initial warming and sea-ice loss, contributed to the amplitude of the deglacial warming and subsequent early Holocene temperatures through a strong positive feedback on the climate system (Past Interglacials Working Group, 2016).

**5.2.2 Holocene**

The warmth during the early Holocene (12-8.7 ka BP) in the southern high latitudes following the Younger Dryas warming can be attributed to high spring insolation causing a longer summer season (Shevenell et al., 2011; Etourneau et al., 2013), which would have enhanced the duration of summer warming and reduced the duration of winter cooling of the open ocean. Insolation-driven changes to the SWW may have also influenced the SST. Reconstructions support that the SWW were weaker generally between 11 and 7 ka BP (Fletcher and Moreno, 2011; Saunders et al., 2018; Figure 8C), shifted southward (Lamy et al., 2010) and that atmospheric circulation was more meridional (Mayewski et al., 2013). The more meridional circulation may have led to a greater poleward penetration of warm air masses and therefore warming over the Southern Ocean. A general weakening of the SWW may have reduced the amount of northward Ekman transport of cold water (e.g. Hall and Visbeck, 2002; Lovenduski and Gruber, 2005), again leading to warming further north. However at the latitude of the Conrad Rise (54°S) it is likely the SWW strengthened, as the southward SWW caused stronger winds in South America at c.53°S (Lamy et al., 2010), and this may have potentially increased heat loss from the ocean surface, damping the warming.

At ca. 8.2 ka BP the SiZer analysis indicated that there was a significant cool interval in the COR1GC record (Figure 5), while the diatom record by Katsuki et al. (2012) from the Conrad Rise also showed a peak in cold species at 8 ka BP. The duration and timing of the SST cooling observed in the COR1GC record (from 8.5-7.9 ka BP) coincides with an AMOC reduction and North Atlantic cooling associated with the 8.2 event (Ellison et al., 2006). The AMOC reduction should have caused a warming of the Southern Ocean via the bipolar seesaw. However a modelling study suggests that cooler NADW was formed in the North Atlantic, due to the retreat of the Laurentide Ice Sheet, and this upwelled in the Southern Ocean causing a cooling in the POOZ between 9 and 7 ka BP (Renssen et al., 2010). The 8.2 event has not generally been observed in records from the southern hemisphere (Alley and Ágústsdóttir, 2005). While there are some indications of a longer period with lower temperatures in Antarctic ice core records at this time (Figure 7 E and F; EPICA Community Members, 2006; Jouzel et al., 2007) and evidence of colder SST at the Antarctic Peninsula (Etourneau et al., 2019), the few other high resolution SST records from the open ocean do not show a cool event (Figure 7; TN057-13PC4; TN057-17TC; Nielsen et al. 2004; Anderson et al., 2009; Divine et al., 2010). Therefore although there appears to have been a cooling on the Conrad Rise at this time it cannot be reliably associated with the 8.2 ka event in the northern hemisphere without additional evidence from other high resolution marine sediment records.

From the mid to late Holocene the spring southern high latitude insolation decreased (Laskar et al., 2004), causing a shorter summer duration and cooling, which is coherent with the northward shift of the SWW (Lamy et al., 2001). Together

this can explain the slight cooling in the COR1GC record, the increased frequency of cold events and increase in sea ice (Figure 4). The winter and summer latitudinal insolation gradient also increased (Laskar et al., 2004; Divine et al., 2010) causing the SWW to strengthen (e.g. Saunders et al., 2018), which may have caused cooling by promoting enhanced northward Ekman transport of cooler water from the south towards the Conrad Rise (e.g. Hall and Visbeck, 2002; Lovenduski and Gruber, 2005).
The strengthened winds may also have increased upwelling and therefore productivity, however at the Conrad Rise this does not seem to have occurred as the estimated diatom flux decreased, particularly after 5 ka BP (Figure 3). Potentially, a strengthening of the SWW to the north, rather than over the Conrad Rise, may explain why upwelling and productivity did not increase, which agrees with evidence from the Atlantic sector that during the mid to late Holocene productivity decreased south of the Polar Front (Anderson et al., 2009).

The COR1GC SST record also shows increasingly variable SSTs through the mid to late Holocene after 8 ka BP (Figure 4). Another record from the Conrad Rise found an increased frequency of cold events from 5.5 ka BP onwards (Katsuki et al., 2012) and the closest ice core record, EDML, shows increasing variability in the $\delta^{18}O$ record from c.6 ka BP onwards (EPICA Community Members, 2006; Fischer et al., 2007; Figure 7E). Spectral analysis conducted on the COR1GC SST record identifies significant quasi-periodicities of ~220 to 260 years (Figure 6) during the last 12,000 years. Similar quasi-periodicities
of 200-300 years have previously been observed in Southern Ocean palaeoceanographic records, including those reflecting SST, sea ice and productivity changes (Leventer et al., 1996; Bárcena et al., 1998; Nielsen et al., 2004; Crosta et al., 2007). Furthermore, quasi-periodicities of 250 years have been identified in a record of westerly wind intensity from the Falkland Islands (Turney et al., 2016) and multi-centennial variations were similarly identified in a 3000 year reconstruction of the SAM (Moreno et al., 2014). These similar patterns of cyclicity in records reflecting both ocean and atmospheric conditions suggest
that they were caused by 1) internal variability and ocean-atmosphere coupling, or 2) ocean and atmosphere responses to a shared forcing. In support of the first hypothesis, model simulations have shown similar 200-300 year oscillations in Southern Ocean temperature and atmospheric circulation (Latif et al., 2013). In a simulation by the Kiel Climate Model, centennial episodes of strong (weak) deep convection in the Weddell Sea caused warm (cold) SSTs and reduced (expanded) sea ice, which altered atmospheric pressure patterns and led to weakened (strengthened) atmospheric circulation (Latif et al., 2013).
In this case, internal oscillations in the climate system have a hemispheric impact, which may explain the similar variability observed in the COR1GC record and other records. In support of the second hypothesis, external forcing, such as cycles in solar activity, may have caused these changes (Leventer et al., 1996; Bárcena et al., 1998; Crosta et al., 2007; Turney et al., 2016), with changes in solar irradiance directly altering ocean SST, and potentially regulating the SAM and associated changes in atmospheric and oceanic circulation, as suggested by some modern evidence that shows a connection between the SAM and
the 11-year solar cycle (e.g. Kuroda and Kodera, 2005; Kuroda, 2018). While it is yet not possible to establish the cause of this cycle, the enhanced variability in the mid-late Holocene appears to reflect the establishment of modern coupled ocean-atmosphere relationships over the Southern Ocean.

**6 Conclusions**

The COR1GC reconstruction of SST variability at the Conrad Rise in the west Indian Sector of the Southern Ocean spans the period from 14.2 to 1 ka BP with an average resolution of ~60 years. SST's were cool (~2.9°C) from 14.2-12.9 ka BP during the ACR before temperatures increased to 5°C between 12.9 and 11.6 ka BP during the Younger Dryas. During the Holocene SST's were more stable and marginally warmer between 12 and 9 ka BP, before a significant, centennial-scale~~sharp~~ cooling event occurred at c.8.2 ka BP, followed by greater SST variability ~~and slightly cooler conditions~~ through the period 8-1 ka BP.

The timing and duration of warming in the COR1GC record at 12.9-11.6 ka BP reflects warming also shown by a precisely dated marine record of SST from the Pacific (Siani et al., 2013) and Antarctic ice core temperature records, supporting that atmospheric and ocean temperatures varied in phase. The COR1GC warming also coincides with palaeo-evidence for a weakened AMOC, rising $CO_2$ levels, greater upwelling in the Atlantic sector of the Southern Ocean, southward displacement and strengthening of the SWW and reduced sea ice, each of which may have contributed to warming in the Southern Ocean. The results support that the warming was initiated by an interhemispheric oceanic and atmospheric reorganisation during the Younger Dryas but does not allow attribution of the relative importance of each process to warming.

For the Holocene period, the most striking change in the COR1GC record is the switch from stable and slightly warmer conditions to variable and slightly cooler conditions at 8.7 ka BP, potentially linked with a global reorganisation of the climate system. It is suggested that the early Holocene warmth may have resulted from higher spring insolation increasing the duration of heat accumulation in the Southern Ocean during the spring-summer season, or as a result of changes to latitudinal heat transport as the SWW weakened. During the mid to late Holocene the enhanced variability may represent the establishment of modern coupled atmosphere-ocean relationships, whereby the SAM and SSTs in the Southern Ocean vary in phase over centennial timescales. This is supported by the identification of a quasi-periodic oscillation of ~200-260 years in the Holocene section of the COR1GC SST reconstruction, which mirrors similar centennial variability identified in reconstructions of the SAM and westerly winds.

**Code/data availability**

The completed dataset can be obtained at pangaea.de: https://doi.pangaea.de/10.1594/PANGAEA.913621

**Author Contributions**

LCO performed the diatom analysis and wrote the manuscript. XC produced the SST record, did training and writing/editing of the manuscript. DD did SiZer analysis. AM, DD, KH, EI, RM edited the manuscript. AM, KH, EI, XC, RM, DD conceptualised and supervised the project. LW and OT conducted laboratory analysis. MI provided core material.

**Competing interests**

The authors declare that they have no conflict of interest.

**Acknowledgments**

This work was funded by the Research Council of Norway (grant no. 248776/E10) and Ministry of Earth Science, Earth System Science Organization (MoES/Indo-Nor/PS-2/2015), through the OCTEL project. We would like to thank Svetlana Divina and Linda Rossignol-Malaizé for preparation of the foraminifera samples for radiocarbon dating. Gravity core sampling and subsample preparation were partly supported by JSPS KAKENHI (grant no. 23244102, 17H06318). RM would like to thank the Director, NCPOR, India for the support to the project and this is NCPOR Contribution No. J-18/2020-21.

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

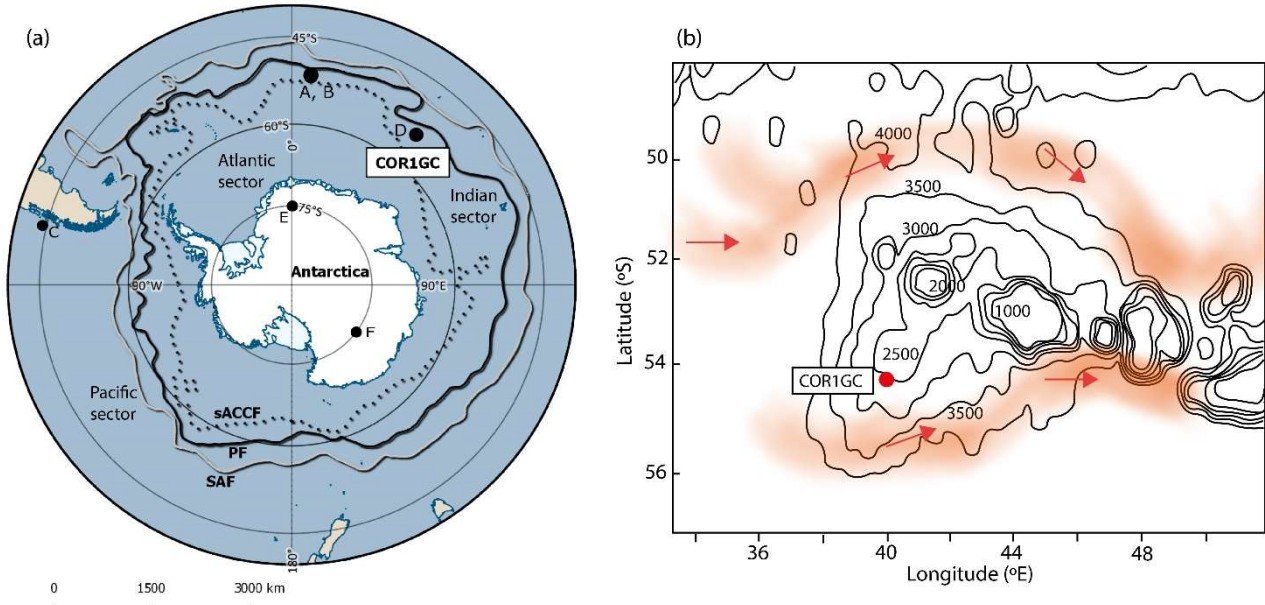

**Figure 1: Location of core COR1GC in the western Indian sector of the Southern Ocean. a) Map of the Southern Ocean and Antarctica. Key terrestrial ice and marine sediment cores mentioned in the text and in Figure 7 are labelled: A) TN057-13-PC4 (Anderson et al., 2009; Divine et al., 2010), B) TN057-17-PC1 (Nielsen et al., 2004), C) MD07-3088 (Siani et al., 2013), D) COR1GC (this study), E) EDML (EPICA Community Members, 2006), F) EDC (Jouzel et al., 2007). The mean location of the southern Antarctic Circumpolar Current Front (sACCF; dotted line), the Polar Front (PF; black line) and Subantarctic Front (SAF; grey line) according to Orsi et al. (1995) are shown. b) Bathymetric map of the Conrad Rise adapted from Ansorge et al. (2008) including areas of high absolute geostrophic velocity in red shading and direction of flow shown by red arrows.**

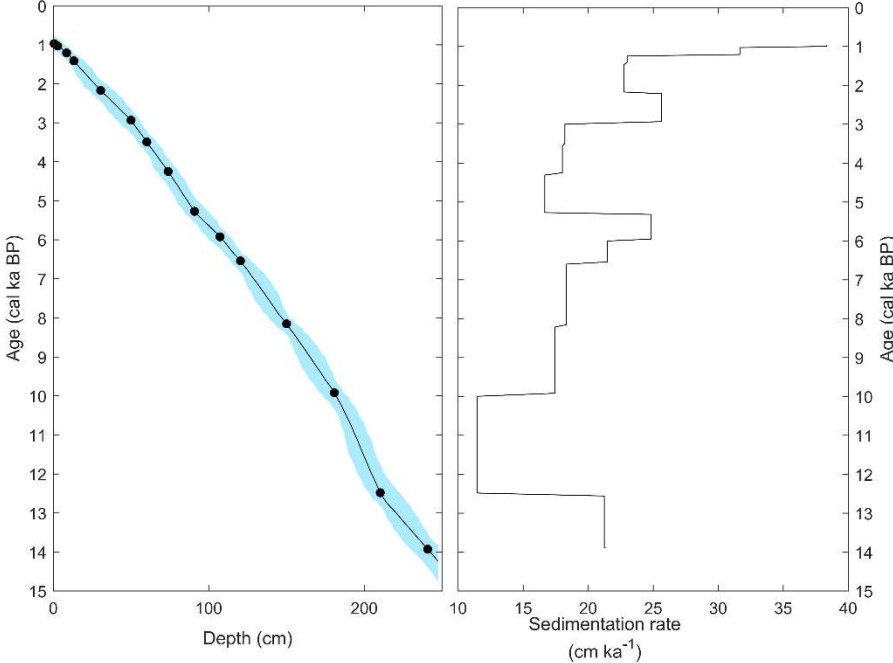

**Figure 2: Left: Age-depth model for core COR1GC. The age model is constrained by 15 calibrated radiocarbon dates, the continuous line is the median age and shaded area represents the 95% confidence interval of the modelled core chronology. Right: Sedimentation rate estimates for COR1GC based on median modelled ages for AMS ¹⁴C dates.**


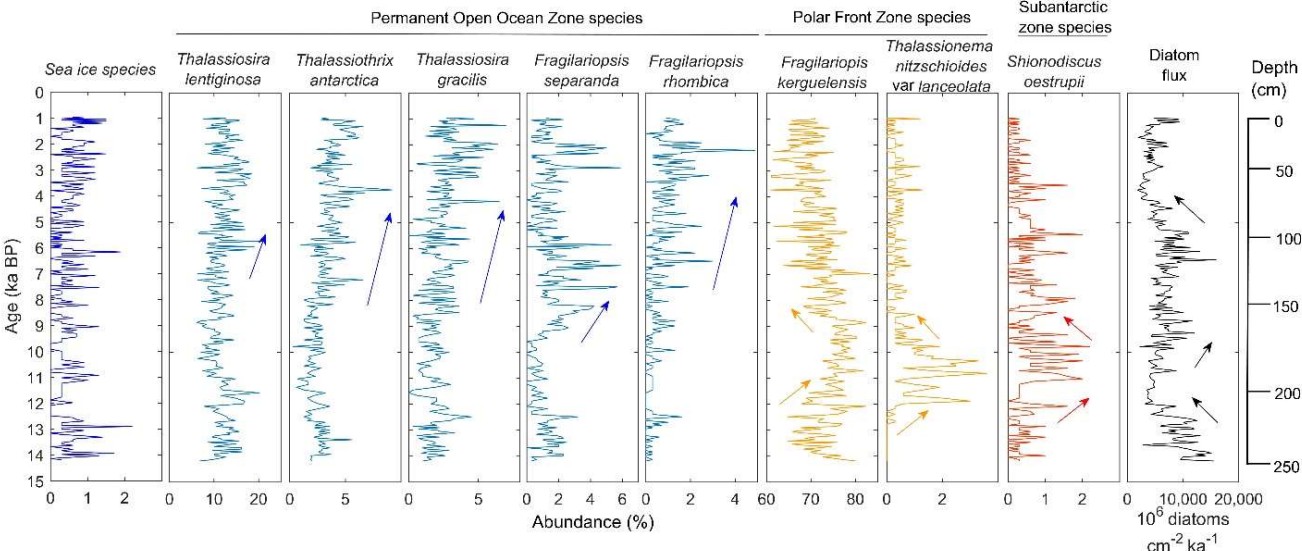

**Figure 3: COR1GC diatom abundances as percentages of total sample and the total diatom flux. Note that the species are plotted on different x-axis scales.**


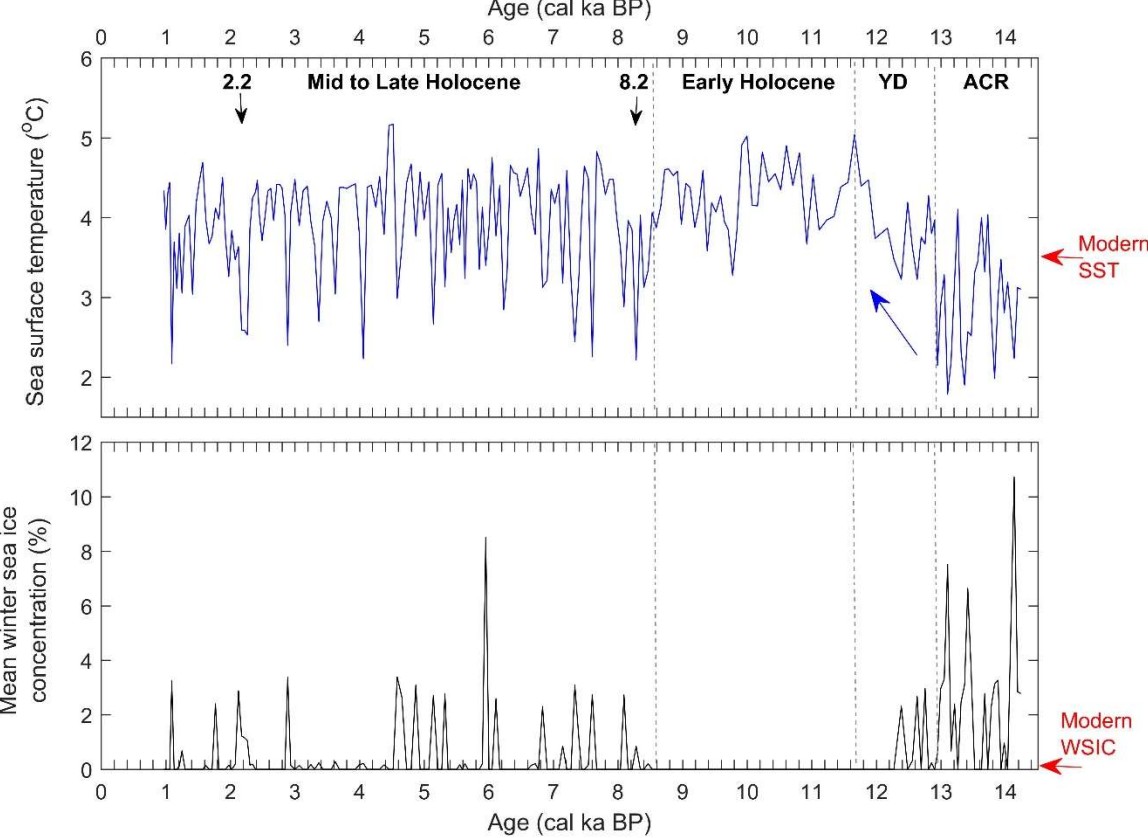

**Figure 4: Reconstructed mean summer (January-March) sea surface temperature and mean winter (September) sea ice**
**concentrations for core COR1GC. Key periods discussed are highlighted, along with significant centennial cool events at c.8.2 and 2.2 ka BP, as identified by the SiZer analysis (Figure 5). The modern average summer SST at 10 m depth is 3.5°C (World Ocean Atlas, 2013) is marked on the graph, while the modern winter sea ice concentration (WSIC) is marked as 0 as the sea ice limit is at 59°S (Park et al., 1998; Parkinson and Cavalieri, 2012).**

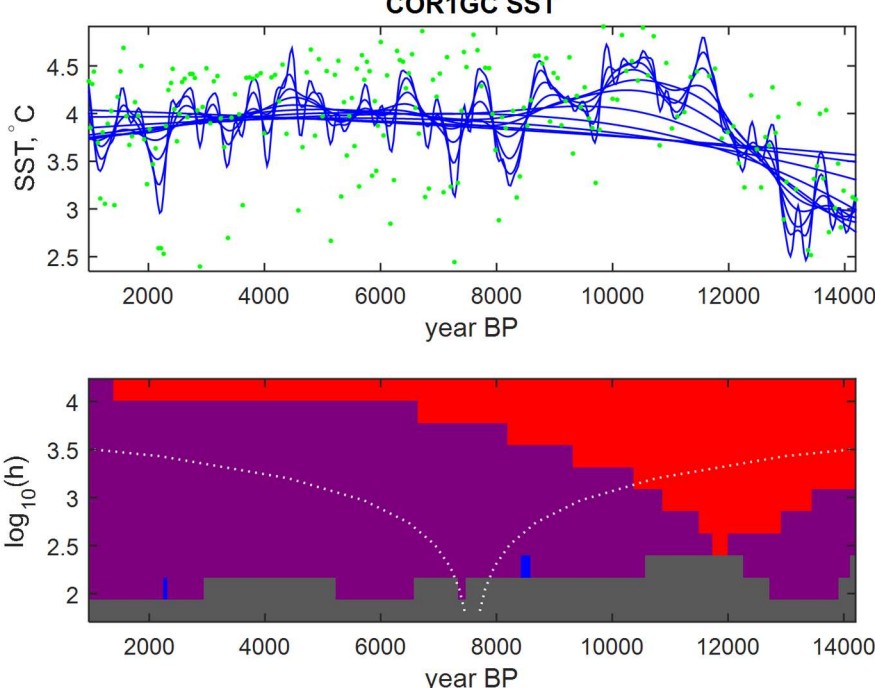


**Figure 5: SiZer analysis of the COR1GC SST reconstruction. Upper panel: Family plot of the reconstructed SST. The green dots represent the raw data. Blue lines show the family of smoothings obtained by the local linear kernel estimator for a range of bandwidths (scales h). Bottom panel: A SiZer map, given as a function of location (time t) and scale h. A significantly positive (increased SST) derivative is flagged as red while a significantly negative (decreased SST) derivative is flagged as blue. The color purple is used at locations where the derivative is not significantly different from zero. The color dark gray is used to indicate that too few data points are available to do a correct inference. The distance between the two dotted lines in the cone-shaped curve for a horizontal line in the SiZer plot can be interpreted as the scale for that level of resolution.**


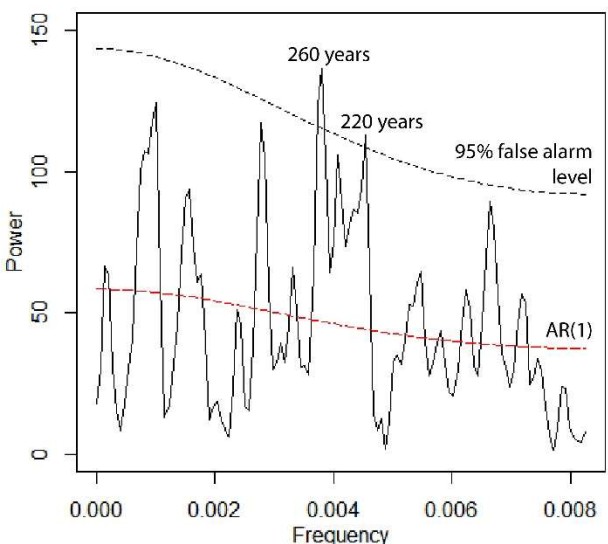

**Figure 6: REDFIT spectral analysis of the COR1GC SST reconstruction during the period 12-1 ka BP. The 95% significance level (Chi2) is shown by the dashed black line. The time series is fitted to an AR(1) red noise model (dashed red line).**

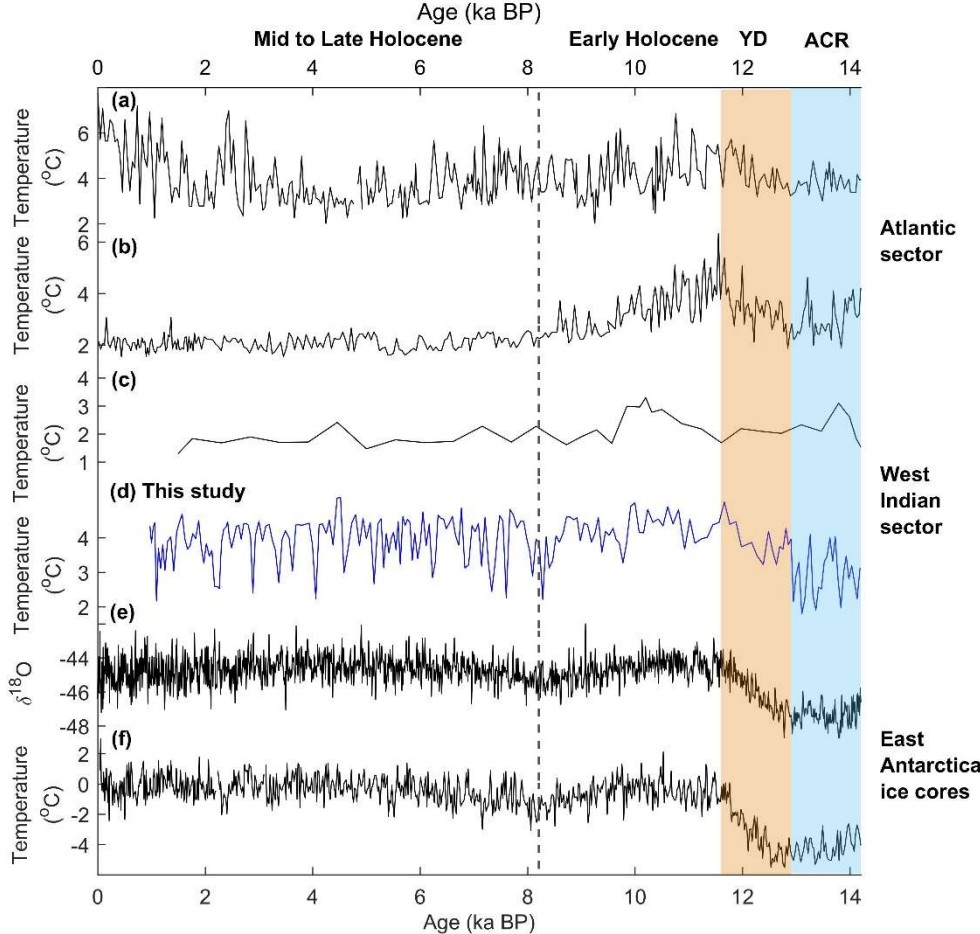

Figure 7: Reconstructions reflecting temperature since 14.2 ka BP from the Southern Ocean and Antarctica at locations shown in Figure 1. A) TN057-17-PC1 diatom-based SST reconstruction (Nielsen et al., 2004), B) TN057-13-PC4 diatom-based SST reconstruction (Anderson et al., 2009; Divine et al., 2010), C) PS2606-6 diatom-based SST reconstruction from the Conrad Rise (Xiao et al., 2016), D) COR1GC SST reconstruction (this study) from the Conrad Rise, E) EDML ice core $\delta^{18}$O reconstruction (EPICA Community Members, 2006), F) EDC ice core temperature reconstruction (Jouzel et al., 2007).

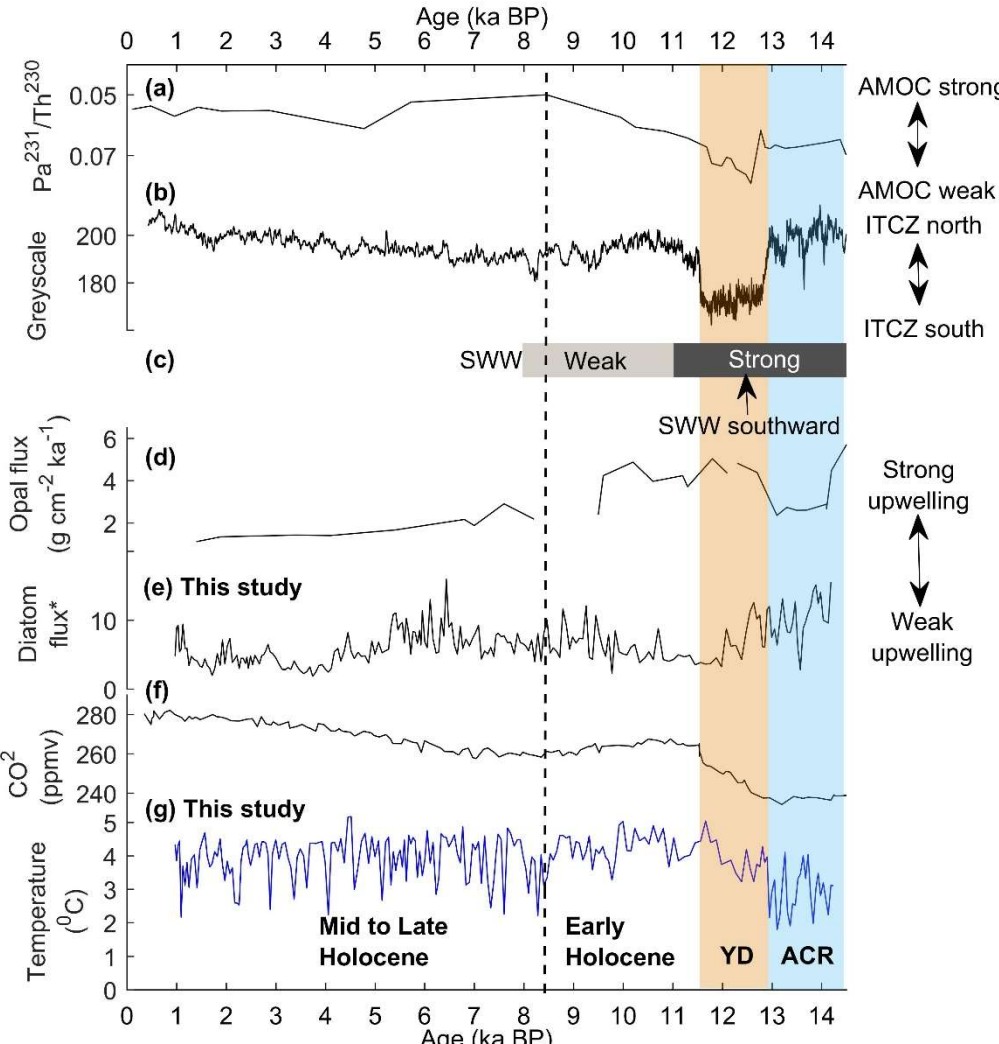

**Figure 8: Summary of key reconstructions showing the late deglacial to Holocene variations in the AMOC, atmospheric circulation and Southern Ocean. A)** AMOC reconstruction based on $Pa^{231}/Th^{230}$ from a core from the western subtropical North Atlantic (McManus et al., 2004); **B)** greyscale measurements from cores from the Cariaco Basin related to productivity, interpreted as reflecting ITCZ position (Hughen et al., 1996); **C)** summary of reconstructed SWW strength based on a collation of palaeoclimate data from the southern mid latitudes (41–52°S; Fletcher and Moreno, 2011); **D)** upwelling in Atlantic sector of Southern Ocean (core TN057-13PC4) based on opal flux (Anderson et al., 2009; Divine et al., 2010); **E)** productivity at the Conrad Rise inferred from the diatom flux in COR1GC, *measured in units: $10^9$ diatoms $cm^{-2}$ $ka^{-1}$; **F)** EDC $CO_2$ reconstruction (Monnin et al., 2001); **G)** COR1GC reconstruction of SST (this study).

**Table 1: AMS radiocarbon dates and calibrated age estimates.**

| Laboratory code | Depth (cm) | $^{14}$C age (years BP) | Age error (years) | Calibrated median age and 95% confidence age range (calibrated years BP) |
|---|---|---|---|---|
| ETH-80023/24 | 0.6 | 2003 | 60 | 970 (770 – 1160) |
| BETA-485984 | 2.9 | 1910 | 30 | 1030 (850 – 1200) |
| BETA-485985 | 8.6 | 2040 | 30 | 1210 (104 – 1390) |
| ETH-80025/26 | 13.2 | 2370 | 65 | 1410 (1210 – 1640) |
| BETA-488823 | 30.5 | 3000 | 30 | 2170 (1900 – 2460) |
| ETH-80027/28 | 50 | 3521 | 65 | 2930 (2640 – 3280) |
| BETA-488824 | 60.2 | 4070 | 30 | 3490 (3200 – 3800) |
| ETH-80029/30 | 73.9 | 4634 | 70 | 4250 (3920 – 4640) |
| BETA-488825 | 90.9 | 5480 | 30 | 5270 (4900 - 5560) |
| ETH-80031/60 | 107.3 | 5966 | 70 | 5930 (5670 – 6230) |
| BETA-488826 | 120.4 | 6580 | 30 | 6540 (6290 – 6840) |
| BETA-488827 | 150.1 | 8160 | 30 | 8160 (7860 – 8440) |
| BETA-488828 | 180.8 | 9600 | 40 | 9920 (9530 – 10350) |
| BETA-488829 | 210.3 | 11570 | 40 | 12490 (11740 – 12840) |
| BETA-488830 | 240.9 | 12890 | 40 | 13930 (13530 – 14400) |

730

**Table 2: Summary table of the suggested forcings of the core COR1GC SST reconstruction, as proposed in Section 5.2**

| Time interval | Initial forcing | Feedbacks and processes |
|---|---|---|
| Younger Dryas | Slowdown of the AMOC | **Direct cause of warming:** Heat accumulation in the Southern Ocean **Indirect causes of warming:** i) ITCZ and SWW shifted southwards and SWW strengthened (greater eddy-transport of heat across the ACC) |

| | | ii) Sea ice loss (ice-albedo changes and amplification of warming. This also increased upwelling) |
| :--- | :--- | :--- |
| | | iii) Increased upwelling (greater outgassing of $CO_2$ and atmospheric warming) |
| Holocene millennial changes | Spring insolation and winter and summer insolation gradients | ~~Early Holocene / mid-late Holocene~~<br><br>~~Length of the summer season: Extended/reduced~~<br><br>~~SWW: weaker and meridional / stronger and zonal~~<br><br>    • ~~increased/reduced atmospheric heat transport southwards~~<br><br>~~reduced/increased Ekman transport of cold water northwards~~<br><br>**Early Holocene (slightly warmer)**<br><br>    • Extended summer season<br><br>    • SWW weaker and meridional: increased atmospheric heat transport southwards and reduced Ekman transport of cold water northwards<br><br>**Mid-late Holocene (slightly cooler)**<br><br>    • Shorter summer season<br><br>    • SWW stronger and zonal: reduced atmospheric heat transport southwards and increased Ekman transport of cold water northwards |
| Mid-late Holocene centennial variability (inc. the 200-260 year cycle) | Internal atmosphere-ocean interactions or solar forcing | i) Internal variability and ocean-atmosphere interactions (variable deep convection rates > SST and sea-ice extent > heating of atmosphere > atmospheric circulation)<br><br>ii) Solar variability: impact on atmospheric and oceanic circulation |