# Peer review of "Sea surface temperature in the Indian sector of the Southern Ocean over the Late Glacial and Holocene"

_Climate of the Past, 2020_

## Referee Comment (RC1) · Anonymous Referee #1 · 2 Apr 2020

This paper presents a new high-resolution diatom based sea surface temperature reconstruction over the past ∼14 ka from the modern Permanent Open Ocean Zone between the Antarctic Polar Front and winter sea ice edge in the western Indian Sector of the Southern Ocean. The high-resolution record resolves centennial- to millennial-scale climate variability that enables a detailed comparison to Antarctic ice core records. Complex processes involving reorganization of atmospheric and oceanic circulation, such as $CO_2$ levels, Southern Westerly Winds, and AMOC, have been attributed to the in-phase variation between the marine and ice core records. Periodicities of 200-260 years were identified in the Mid-Late Holocene interval, they were related to high latitude atmospheric circulation and Southern Ocean convection.

[Figure]

This paper provides a novel sea surface temperature record from an area very limited records are available, especially compared to the Atlantic and Pacific sectors of the Southern Ocean. Thus, the new data from this study augment valuable information to a more comprehensive understanding of environmental changes in the Southern Ocean. However, there are a few aspects that the authors can clarify or improve before publication.

1. Study area This part is too extensive. The text can be more concise and focused on information directly related to this study.

2. Methods (a) Age model: Line 117-122: The authors claim that the reservoir applied in the Atlantic sector of the Southern Ocean derived from comparison between 14C ages and 226Ra-in-barite ages (van Beek et al., 2002) are not reliable "because large variations of ∼400 years were observed between consecutive depths", and listed studies showing relatively constant reservoir in the Southern Ocean (Hall et al., 2010; Siani et al., 2013). Regarding this statement, I hold different opinion. The 226Ra-in-Barite ages are consistent and in good order in van Beek et al. (2002). The resulted large variation in calculated reservoir ages are mostly because 14C ages measured in different labs (Kiel & Aarhus). The 14C ages from Aarhus is systematically ca. 300 years younger than those from Kiel, which lead to 300-400 years variation of reservoir changes in consecutive depth. For this reason, a mean value of ca. 1100+-210 years reservoir was taken for mid-late Holocene, and adopted to other South Atlantic cores (e.g., Xiao et al., 2016). As such, the results in van Beek et al. (2002) do not really conflict with a relatively constant reservoir during the mid-late Holocene in the South Atlantic. Besides, in Hall et al. (2010), the authors propose reservoir ages of 1144+-120 years for the mid-late Holocene in the Ross Sea sector of the Southern Ocean, similar to the results from the South Atlantic. In Siani et al. (2013), their record MD07-3088 is located at ∼ 46°S, much north of the modern Subantarctic Front. We could expect different reservoir effect in different water masses, such as in cores south of the Polar Front and north of the Subantarctic Front. It is good that the reservoir effect applied in this paper results in a good alignment between the marine and ice core records. However, due to the lack of knowledge of precise reservoir variation through time, slight phase shifts in marine records can be attributed to age uncertainty.

(b) SST reconstruction Line 132: Do the SSTs reconstructed in this study represent true surface (0 m) temperatures? Because other transfer functions estimate temperatures at 10 m water depth (Esper et al., 2014), which could result in some difference when comparing with other records (e.g., variation amplitude).

3. Discussion (a) Line 216: cores located north of the modern APF generally show a late Holocene warming trend. The referenced cores showing late Holocene warming cited here include TN057-17 (Nielsen et al., 2004; Divine et al., 2010) from the Southern Ocean Atlantic sector close to the modern Polar Front, and MD07-3088 off Chilean margin far north of the Subantarctic Front (Siani et al., 2013). However, there are reconstructions adjacent to these two cores do not show such warming. For example, PS1654/ODP1093 close to TN057-17 do not show clear warming during the late Holocene, although the resolution is much lower in PS1654/ODP1093 in the mid-late Holocene interval (Xiao et al., 2016). Further north, a mid-late Holocene cooling was inferred from ODP1098 and ODP1090 (Xiao et al., 2016). Alkenone-derived SST records off Chilean margin (including MD07-3088) also suggest a mid-late Holocene cooling which is in contrast to the MD07-3088 SST record derived from foraminifera assemblage using modern analogue technique (Lamy et al., 2010, Nat. Geosci.; Haddam et al., 2018, QSR). In general, the majority of the available records show a mid-late Holocene cooling in the Southern Ocean (e.g., Bostock et al., 2013; Xiao et al., 2016). While the late Holocene SSTs reconstructed in core TN057-17 far exceeding modern temperatures (by $\sim$2 °C) (Nielsen et al., 2004; Divine et al., 2010) need more explanation.

(b) Line 291-293: Given the timing and synchronicity between CO2 increases and some of the temperature records, it is possible that CO2 caused much of the warming... -> How do you quantify the warming by redistribution of heat between

northern and southern hemispheres during the period of AMOC slow-down (bipolar-seesaw), and the warming due to CO2 rise? I understand the positive feedback of CO2 increase on warming, but I do not understand how does it reach the conclusion that CO2 is the main driver of Southern Ocean warming just based on the synchronicity of warming and CO2 rise.

(c) In chapter 5.2.2, the authors cited several references showing a weakening of SWW during the early Holocene (Fletcher and Moreno, 2011; Saunders et al., 2018). However, it needs to be mentioned that, the weakening of SWW at this time interval was north of modern Polar Front. A southward shift of the SWW was inferred during this time interval (as suggested by opal accumulation, Anderson et al., 2009; Xiao et al., 2016), which indicate the SWW in regions further south can be stronger. Such interpretation is misleading in terms of eddy transport of heat and atmospheric circulation, which occurred also in the following text. The core latitudinal band of the SWW shifts with the warming/cooling of the Southern Hemisphere.

(d) Line 310-311: . . . the few high resolution SST records from the open ocean do not show a cooling (Figure 7; TN057-13PC4; TN057-17TC). . . -> Do you mean a cooling event? Because TN057-13, together with many other South Atlantic cores south of the modern Polar Front, show persistent cooling after ca. 8 ka (Divine et al., 2010; Xiao et al., 2016). In fact, the SiZer analysis of SST (Fig. 5) and cold water species (Fig. 3) show many cooling episodes during the mid-late Holocene, with amplitude similar or even steeper than that around 8 ka. It may reflect millennial-scale climate variability during the mid-late Holocene, rather than linking to a single cooling event in the Northern Hemisphere.

(e) The 220-260 yrs periodicity found in the SST record needs more explanation. The authors listed a number of published records around Antarctica with similar cyclicities, that most of them were related to solar activities in their respective publications. The authors then introduced other studies linked such cyclicity to atmospheric circulation such as SAM and ENSO, and rapid climatic events globally. This part is confusing

as it seems related to so many processes and how exactly they work needs better elaboration.

Some minor points are as follows:

Line 34: For example during . . . -> For example, during. . . Younger Dryas (13.02-11.76 ka BP) -> 12.9-11.7 ka (Rasmussen et al., 2014 Quat. Sci. Rev. 106, 14-28)

Line 35: accumulate in the South Atlantic -> accumulate in the Southern Ocean causing a "bipolar seesaw" characterized by. . . -> causing a "bipolar seesaw" pattern characterized by. . .

Line 63: during the Holocene there is. . . -> during the Holocene, there is. . .

Line 71-72: extending from the Subantarctic Front in the north. . . -> extending from the Polar Front in the north. . . (Diekmann, 2007, Deep-Sea Res. II 54, 2350-2366)

Line 74: (Park et al., 1998) -> Park et al. (1998) is not a sea ice study. This can be replaced by more recent satellite observations of sea ice extent, such as Parkinson & Cavalieri, 2012, The Cryosphere 6, 871-880.

Line 101-102: . . .in the southwestern Indian sector of the Southern Ocean. . . -> To make the text more concise, this can be removed as there is an extensive description of the study region in the previous chapter.

Line 108: Neogloboquadrina pachyderma -> species name should be italic.

Line 134: Winter sea ice concentration was. . . -> abbreviation (WSIC) should be noted here

Line 173: A number of references describing environmental preferences of the species need to be mentioned before referring certain species to, e.g., PFZ species, POOZ species, and sea ice species. References can be cited are Zielinski and Gersonde, 1997, Palaeo 3; Crosta et al., 2005, Palaeo 3; Armand et al., 2005, Palaeo 3; Romero et al., 2005, Palaeo 3; Esper et al., 2010, Palaeo 3.

Line 234: This record was constrained. . . -> The MD07-3088 record. . .

Line 248: warming in the COR1GC and ice core records. . . -> warming in the COR1GC and Antarctic ice core records. . .

Line 262: Hogg et al., 2007 -> Hogg et al., 2008

Line 299: Reconstructions support that the SWW were weaker between 11 and 7 ka BP. . . -> The authors need to mention that the weakening of SWW at this time interval was north of modern Polar Front. A southward shift of the SWW was inferred during this time interval (as suggested by opal accumulation, Anderson et al., 2009; Xiao et al., 2016), which indicate the SWW in regions further south can be stronger.

Line 303-304: This is uncertain as modern observations instead show that weaker SWW's cause reduced eddy activity and less poleward heat transport across the ACC, resulting in cooling (Hogg et al., 2007; Screen et al., 2009). -> Reference error: Hogg et al., 2008 As above, SWW did not weaken at all latitudinal bands.

Line 306-309: The duration and timing of the SST cooling observed in the COR1GC record coincides with an AMOC reduction and North Atlantic cooling associated with the 8.2 event (Ellison et al., 2006) however it has not generally been observed in records from the southern hemisphere (Alley and Ágústsdóttir, 2005). -> Modeling study by Renssen et al. (2010) suggests the upwelling of cooler NADW in the Southern Ocean could result in the drop of surface temperature between 9 and 7 ka.

Line 315: From the mid to late Holocene the high latitude insolation decreased (Divine et al., 2010) -> Unclear expression. Do you mean southern high latitude? Summer insolation increased during the mid-late Holocene in southern high latitude. Do you mean winter or spring insolation? or annual insolation? Divine et al., 2010 is not the proper reference for insolation, use Laskar et al., 2004 instead.

Line 315-316: causing the SWW's to shift northward -> coherent with the northward shift of the SWW

Line 317: Divine et al., 2010 -> use Laskar et al., 2004 for insolation reference

Line 317: causing the SWW's to strengthen (e.g. Saunders et al., 2018) -> as above, need to mention strengthening in areas north of the modern Polar Front. It can be weakening in the south.

Line 319-320: The strengthened winds may also have increased upwelling and therefore productivity. -> south of the Polar Front, upwelling was reduced during the mid-late Holocene (Anderson et al., 2009).

Line 336: in the atmosphere may. . . -> in the atmosphere circulation

Line 356: spring insolation extending the summer season -> what does it mean?

Fig. 2: as the last age control point is at 240.9 cm (13.93 ka BP), why there is a sudden increase in sedimentation at the core base, where no actual age constraint is available?

Fig. 3: I would suggest to arrange the species from left to right according to their ecological preferences in terms of temperature. For example, F. separanda, F. rhombica and T. gracilis are cold water species, which can be placed next to the sea ice species. Such arrangement will show evolution pattern of the diatom assemblage more clearly, which reflect environment changes.

Fig. 8: the opal flux record in panel (d) seems incomplete? There are breaks in the curve.

---

## Referee Comment (RC2) · Anonymous Referee #2 · 15 Apr 2020

Title: Sea surface temperature in the Indian sector of the Southern Ocean over the Late Glacial and Holocene Authors: Lisa C. Orme et al. MS No.: cp-2020-23 MS Type: Research article

Overview and general recommendation

The sequence of warmings and coolings associated with the last deglaciation and the Holocene has shown contrasting patterns between southern and northern high latitudes. Global-scale processes such as variations of AMOC strength and the alteration of atmospheric circulation seem responsible for this contrast. Most of the high-resolution paleorecords studied so far were gained from Antarctic ice cores. Therefore,

acquiring high-resolution proxy records of past sea surface temperature is relevant for finding out how spatio-temporal patterns of water temperature evolved in the Southern Ocean and whether any links with Antarctic temperature variability are recognized.

Orme and co-authors present a high-resolution, diatom-based record of sea-surface temperature gained in the western Indian sector of the Southern Ocean (Core KH-10-7 COR1GC, ca. $54.27°$S, $39.77°$E, WD 2834 m). Their sediment record spans the past 14.2 Ka BP. The age model bases on fifteen AMS radiocarbon dates obtained on mono-specific samples of the planktic foraminifera Neogloboquadrina pachyderma sin. The average temporal resolution of their diatom counts is ca. 60 years (diatom analysis was conducted every cm in the 2.48 m gravity core). By using the Modern Analogue Technique applied to diatom assemblages, Orme and colleagues estimate the summer SST and the winter sea ice concentration. They describe and discuss patterns, timing and magnitude of sea-surface temperature variability through the late deglaciation and the Holocene in the western Indian sector of the Southern Ocean and possible links to global and regional forcings and mechanisms.

The Introduction presents basic information on (1) deglacial events and Holocene intervals, and (2) main mechanisms/forcings behind temperature in the Southern Ocean; it reads well and helps the reader less familiar with issues addressed later in the MS. The Methodology is clearly written. Results are concisely presented; the results representation however can improve (see suggestions below). Figures are self-explanatory and necessary in number. References are satisfactory.

A major concern is how the Discussion is organized. Throughout the Discussion, there are several inconsistencies and several vague statements that lack scientific support. Some ideas are shortly presented, without any further and deeper discussion. Too many forcings and mechanisms are offered as possible explanations for the SST variations (solar forcing, internal climate variability, sea-ice and productivity changes, ocean-atmosphere coupling, rapid climate change events globally, establishment of modem ENSO amplitude and frequency), without clearly distinguishing which

mechanism/forcing/s was/were more important when, and the reader gets lost. Please consider: (1) shortening and focusing the discussion, and (2) adding a Table summarizing with main mechanism/forcing/s for each for each of the discussed intervals.

Below I list several minor comments and give some suggestions which might be helpful to improve your MS: L. 66-67: this is repeated several times throughout the Intro. Please revise. L. 93-94: since the authors state that 'Topography has a strong influence on the position and form od the ACC in this region', they should provide a more detailed figure of the study area in Fig. 1, including bathymetric information. L: 108: Neogloboquadrina pachyderma should be in italics. L. 115-125: Using reservoir age from a core gained in a mid-latitude coastal upwelling system is -at the least-risky and caution is advised. Oceanographical and nutrient conditions in the SE Pacific Ocean are quite different from those in the Conrad Rise and can hardly be straightforwardly applied. The high-resolution sampling (every cm) make the age model uncertainties even larger. Additionally, reworking should be considered/discussed. L. 145-46: these three processes strongly impact your diatom signal (the basis of your SST reconstructions), but it is hardly discussed in 5. Discussion. L. 172: please give age ranges for the Holocene (which age definition of Holocene did you follow?) I suggest adding a box in the upper part of Fig 4, indicating the main intervals of the last deglaciation (YD, ACR, etc.) and Holocene (early/middle/late). This is presented later in Fig 7, but it should be earlier when Results are described. L. 173-183: References for the paleoecological information of the diatom species should be provided here. The reader does not know where the species ecology does come from. L. 184: 'The estimated total diatom abundance shows a decreasing trend through the record', please revise this statement. It is not quite correct to state that the total diatom abundance (TDA) shows an overall decreasing between last deglaciation and the latest Holocene. Indeed, TDA varies strongly up to 5.5 Ka BP and experienced afterward a two-step decrease, first around 5.5 Ka BP and later between 4 and 1 ka BP. L. 190-198: all short intervals mentioned here should be easily recognizable in Fig 4. Please add some arrows to help the reader to better understand what you are trying to communicate. Moreover, add marks between millennial ages in Fig 4: your Results description goes into centennial-scale description (e.g., Between 11.6 and 8.7 ka BBT, etc). Without these centennial-scale marks is even more difficult to recognize whichever trends and shifts occurred. L. 210: 'high temperatures between 11.6 and 8.7 ka BP during the Early Holocene, followed by a cooling trend thereafter': this is a matter of interpretation. The range of SST variability is larger (larger amplitude) between 8 and 1 ka BP than earlier between 8.2 and 11.8 ka BP. However, is it correct to state that a cooling occurred during the middle to late Holocene? I am not able to recognize a clear decreasing trend in your data depicted in Fig. 7d. L. 215: 'Although most records, including COR1GC, show a long-term cooling over the Holocene (Xiao et al., 2016)'. Please revise: the PS2606-6 SST record shows similar values during YD and the entire Holocene. SO, where is the Holocene cooling? L. 218: 'The records from Bouvet Islands', where is this? Which sector of the SO? Please provide more accurate information. L. 222-23: 'Our new record from core COR1GC conversely shows SSTs were 1°C lower during the ACR compared to the mid-late Holocene', Is this 1°C difference statistically significant? 1°C of SST difference lays surely within the range of variability of your SST reconstruction. L. 233: can a 2-3°C rise of SST during the Holocene -compared to last deglaciation- as WARM conditions? I understand that it was warmer, but it is not a warm environment per se, mainly when your SST reconstructions is compared with records from mid and low latitudes. L. 241-43: I agree with these mechanisms and forcings impacting the reconstructed SST record at your core site. However, since diatoms experience dissolution between sea surface and the bottom of the ocean, I can assume that your reconstructed SST values vary depending on which species did it to the sediment. There is, however, no discussion on the possible role of preferential dissolution/preservation of diatoms (see also l. 145-46). L. 245: 'Southern high latitude warming (Termination 1b) during the Younger Dryas', this is given as one unique interval before (see l. 209-10). Please revise and rephrase correspondingly. L. 266-67: 'Greater upwelling has been shown by higher opal deposition to the south of the Polar Front in the Southern Ocean (Atlantic, Indian and Pacific sectors) through the period

12.7- 11.5 ka BP', this is true. However, your TDA data do not show any significant difference among ACR, YD, and early Holocene. Therefore, your data offer no convincing evidence of an intensification of upwelling following the last deglaciation. L. 275-76: 'as tentatively inferred from the slightly increased diatom abundances at 12.7-12 ka BP'. This is hardly recognizable in your COR1GC record. Your TDA does not actually differ from earlier and later values. Please revise. L. 283-84: 'Indeed a southward shift is indicated by an increase in Polar Front species at c. 12 ka BP', you mean Thalassionema nitzschioides var. nitzschioides? The increase is not that clear in F. kerguelensis. L. 296: 'has been attributed to high annual, winter and spring insolation levels', please clarify: do you mean average annual insolation or winter and spring insolation? L. 315: SSW's, misspelling. See also below l. 317. L. 316: 'together can explain the gradual cooling in the COR1GC record', please provide SST range, average and 1STD for your mid and late Holocene SST record. L. 320: 'however it is not clear if this occurred as although there was a gradual increase in sedimentation rate, potentially reflecting an increased deposition of diatoms'. Be cautious with this: according to your data, no increase in total diatom abundance occurred at this time. L. 326-340: this is a quite different story from all the above discussion and confuses the reader. Several forcings are mentioned/shortly discussed in 14 lines (solar forcing, internal climate variability, sea-ice and productivity changes, ocean-atmosphere coupling, rapid climate change events globally, establishment of modem ENSO amplitude and frequency). Presenting an alternative climate scenario at the very end of the manuscript (rapid climate change events globally and establishment of modem ENSO amplitude and frequency), without any further discussion makes this subsection even more confusing and does not add anything valuable to your overall Discussion.

Figures Fig 1: please consider (1) zooming into the closest area to core COR1GC (include bathymetry), and (2) identifying the Atlantic, Indian and Pacific sectors of the Southern Ocean. In the caption the references for cores TN057-13-PC4, TN057-17-PC1, MD07-3088, EDML, and EDC should be presented. Fig. 3: note that you name core COR1GC differently depending on the figures. Please revise. var. in Thalassionema nitzschioides var. nitzschioides should not be italics. Thalassiosira oestrupii has been renamed for years already, the current name is Shionodiscus oestrupii. Consider using exponential nomenclature for x-axis of TDA. Consider adding some arrows to lead the reader in better understanding major shifts/changes in (1) the species composition of the diatom assemblage, and (2) total diatom abundance. Fig. 4: Please consider adding a box in the upper panel indicating YD, early/mid/late Holocene, etc. (see Fig 7) Consider also adding the (1) average and 1STD of your own data, and (2) present-day summer and winter SSTs., and (3) present-day mean winter sea ice concentration. Fig. 7: the long-term pattern of your SST data is pretty similar to that of East Antarctica cores: low ACR values, increase during the YD, and warmer Holocene SST. A simple statistical analysis should help you to better understand the trends. Your SST record shows ten SST minima (cooling) during the Holocene: it seems to me that most of these minima are made by only ONE sample. This can be part of regular variability of the diatom assemblage and not at all related with actual SST variations. Caution is advised in the interpretation of these minima!

————————————————————

---

## Author Comment (AC1) · 1 May 2020

[please note author responses are shown by '>']

This paper presents a new high-resolution diatom based sea surface temperature reconstruction over the past âĹij14 ka from the modern Permanent Open Ocean Zone between the Antarctic Polar Front and winter sea ice edge in the western Indian Sector of the Southern Ocean. The high-resolution record resolves centennial- to millennial-scale climate variability that enables a detailed comparison to Antarctic ice core records. Complex processes involving reorganization of atmospheric and oceanic circulation, such as $CO_2$ levels, Southern Westerly Winds, and AMOC, have been

attributedtothein-phasevariationbetweenthemarineandicecorerecords. Periodicities of 200-260 years were identified in the Mid-Late Holocene interval, they were related to high latitude atmospheric circulation and Southern Ocean convection. This paper provides a novel sea surface temperature record from an area very limited records are available, especially compared to the Atlantic and Pacific sectors of the Southern Ocean. Thus, the new data from this study augment valuable information to a more comprehensive understanding of environmental changes in the Southern Ocean. However, there are a few aspects that the authors can clarify or improve before publication.

>We would like to thank reviewer 1 for the very helpful comments provided, which we feel have improved the manuscript.

1. Study area. This part is too extensive. The text can be more concise and focused on information directly related to this study.

> We have removed sections of the study area not relevant to the Conrad Rise so that it is shorter and concise.

2. Methods (a) Age model: Line 117-122: The authors claim that the reservoir applied in the Atlantic sector of the Southern Ocean derived from comparison between 14C ages and 226Ra-in-barite ages (van Beek et al., 2002) are not reliable "because large variations ofâĹij400 years were observed between consecutive depths", and listed studies showing relatively constant reservoir in the Southern Ocean (Hall et al., 2010; Siani et al., 2013). Regarding this statement, I hold different opinion. The 226Ra-inBarite ages are consistent and in good order in van Beek et al. (2002). The resulted large variation in calculated reservoir ages are mostly because 14C ages measured in different labs (Kiel & Aarhus). The 14C ages from Aarhus is systematically ca. 300 years younger than those from Kiel, which lead to 300-400 years variation of reservoir changes in consecutive depth. For this reason, a mean value of ca. 1100+-210 years reservoir was taken for mid-late Holocene, and adopted to other South Atlantic cores (e.g., Xiao et al., 2016). As such, the results in van Beek et al. (2002) do not

really conflict with a relatively constant reservoir during the mid-late Holocene in the South Atlantic. Besides, in Hall et al. (2010), the authors propose reservoir ages of 1144+-120 years for the mid-late Holocene in the Ross Sea sector of the Southern Ocean, similar to the results from the South Atlantic. In Siani et al. (2013), their record MD07-3088 is located at âĹij 46◦S, much north of the modern Subantarctic Front. We could expect different reservoir effect in different water masses, such as in cores south of the Polar Front and north of the Subantarctic Front. It is good that the reservoir effect applied in this paper results in a good alignment between the marine and ice core records. However, due to the lack of knowledge of precise reservoir variation through time, slight phase shifts in marine records can be attributed to age uncertainty.

>Thank you for alerting us to the issue of different labs in the van Beek et al (2002) paper, we have removed the sentences about these reservoir ages not being reliable. We have also added that the Siani reservoir calculations come from north of the subantarctic front but also acknowledge that it is bathed in water from the Southern Ocean which is deflected north along the South American continent (see our full response to this in reviewer 2 comment L. 115-125). The paragraph as a whole has been rephrased, so that instead of justifying the use of a constant reservoir age, we instead acknowledge that in other SO sectors there is evidence for some changes. We conclude that a lack of evidence for the Indian sector has guided our decision to use a constant estimate of reservoir age and that as a result there is some age uncertainty particularly in the older part of the record. We hope that this provides an open and balanced acknowledgement of the chronological issues effecting records from the SO. Nevertheless, we feel that the good correlation between the SST record and ice core records over the deglacial supports that our chronology is fairly accurate. See lines 105-119.

(b) SST reconstruction Line 132: Do the SSTs reconstructed in this study represent true surface (0 m) temperatures? Because other transfer functions estimate temperatures at 10 m water depth (Esper et al., 2014), which could result in some difference when comparing with other records (e.g., variation amplitude).

>This reconstruction represents SST at 10 m depth. As there is very little difference in temperature between 0 m and 10 m depth, in the order of 0.05°C when considering the whole database, this can be considered as representative of the true surface temperature. Previous studies that refer to sea-surface temperatures (SST) when reconstructing paleo-temperatures are also calibrated against 10 m depth (Zielinski et al., 1998; Esper et al., 2014; Xiao et al., 2016; etc...), so this cannot account for differences with other records. We have now specified in section 3.3 that the SST represents the 10 m depth and was calibrated against modern 10 m depth temperatures. See lines 122 and 125.

3. Discussion (a) Line 216: cores located north of the modern APF generally show a late Holocene warming trend. The referenced cores showing late Holocene warming cited here include TN057-17 (Nielsen et al., 2004; Divine et al., 2010) from the Southern Ocean Atlantic sector close to the modern Polar Front, and MD07-3088 off Chilean margin far north of the Subantarctic Front (Siani et al., 2013). However, there are reconstructions adjacent to these two cores do not show such warming. For example, PS1654/ODP1093 close to TN057-17 do not show clear warming during the late Holocene, although the resolution is much lower in PS1654/ODP1093 in the mid-late Holocene interval (Xiao et al., 2016). Further north, a mid-late Holocene cooling was inferred from ODP1098 and ODP1090 (Xiao et al., 2016). Alkenone-derived SST records off Chilean margin (including MD07-3088) also suggest a mid-late Holocene cooling which is in contrast to the MD07-3088 SST record derived from foraminifera assemblage using modern analogue technique (Lamy et al., 2010, Nat. Geosci.; Haddam et al., 2018, QSR). In general, the majority of the available records show a midlate Holocene cooling in the Southern Ocean (e.g., Bostock et al., 2013; Xiao et al., 2016). While the late Holocene SSTs reconstructed in core TN057-17 far exceeding modern temperatures (byâĹij2 âǓęC) (Nielsen et al., 2004; Divine et al., 2010) need more explanation.

>Lines ∼225: We have removed the Siani et al (2013) reference from here as it is true

that it is located north of the SAF and also is in the Pacific sector, and as such may not be expected to share the temperature changes observed in the Atlantic or Indian sectors. We have altered the wording so that it no longer states that temperatures warmed north of the PF and cooled to the south but instead states that there is some regionally heterogeneity (line 235). As suggested we have added that the cores nearby to TN057-17 (PS1654/ODP1093) show little change in the mid-late Holocene but acknowledged that they are very low resolution in this Holocene section of the core (lines 232-235). We have not added a detailed discussion of why temperatures in TN057-17 show warming, as we consider this to be outside of the scope of this paper and this section of the paper is about discussing patterns of temperature change, rather than the causes.

(3b) Line 291-293: Given the timing and synchronicity between CO2 increases and some of the temperature records, it is possible that CO2 caused much of the warming... -> How do you quantify the warming by redistribution of heat between northern and southern hemispheres during the period of AMOC slow-down (bipolar seesaw), and the warming due to CO2 rise? I understand the positive feedback of CO2 increase on warming, but I do not understand how does it reach the conclusion that CO2 is the main driver of Southern Ocean warming just based on the synchronicity of warming and CO2 rise.

>It was not our intention to say that CO2 was the only cause, rather to observe that the relative importance of different contributing factors is still not known and that the similarity between the COR1GC SST and CO2 changes indicates a possible link. We have rewritten the section (lines 305-314) to now emphasise that we are questioning the relative importance of factors such as CO2, rather than suggesting that this could be the primary cause. We now state that through positive feedback processes in response to initial warming CO2 likely contributed to the amplitude of the warming trend.

(3c)In chapter 5.2.2, the authors cited several references showing a weakening of SWW during the early Holocene (Fletcher and Moreno, 2011; Saunders et al., 2018).

However, it needs to be mentioned that, the weakening of SWW at this time interval was north of modern Polar Front. A southward shift of the SWW was inferred during this time interval (as suggested by opal accumulation, Anderson et al., 2009; Xiao et al., 2016), which indicate the SWW in regions further south can be stronger. Such interpretation is misleading in terms of eddy transport of heat and atmospheric circulation, which occurred also in the following text. The core latitudinal band of the SWW shifts with the warming/cooling of the Southern Hemisphere.

>We have re-written the paragraph from lines 315-325 to now consider the SWW shifts and slightly alter the interpretation. In Lamy et al (2010) the sites at 53°S show the SWW's were stronger in the early Holocene, and therefore at the Conrad Rise at 54°S there were also likely stronger winds as you suggest, despite an overall weakening of the SWW. We now observed that the generally weaker winds more widely likely reduced the northward Ekman transport of cold water from the south resulting in warming (e.g. Hall and Visbeck, 2002; Lovenduski and Gruber, 2005) and warmer air may have more frequently crossed the area due to the meridional circulation. However we also now note that as there were likely stronger winds over the Conrad Rise (Lamy et al., 2010) as you suggest, this potentially increased the heat loss from the ocean to the atmosphere damping the warming. In the conclusions (at lines 389-391) we have also adjusted it so it now reads: 'It is suggested that the early Holocene warmth may have resulted from spring insolation increasing the heat accumulation in the Southern Ocean during the spring-summer season, or as a result of changes to latitudinal heat transport as the SWW weakened.' This acknowledges both the increased southward heat transport from the meridional atmospheric circulation and also the reduced cold water from Ekman transport.

(3d) Line 310-311: ... the few high resolution SST records from the open ocean do not show a cooling (Figure 7; TN057-13PC4; TN057-17TC)... -> Do you mean a cooling event? Because TN057-13, together with many other South Atlantic cores south of the modern Polar Front, show persistent cooling after ca. 8 ka (Divine et al., 2010;

Xiao et al., 2016). In fact, the SiZer analysis of SST (Fig. 5) and cold water species (Fig. 3) show many cooling episodes during the mid-late Holocene, with amplitude similar or even steeper than that around 8 ka. It may reflect millennial-scale climate variability during the mid-late Holocene, rather than linking to a single cooling event in the Northern Hemisphere.

>We have corrected the wording to make our meaning more clear, at line 335. We meant a cool event rather than a persistent cooling. As we discuss the late Holocene high SST variability in the following paragraphs we have not added anything about this here.

(3e) The 220-260 yrs periodicity found in the SST record needs more explanation. The authors listed a number of published records around Antarctica with similar cyclicities, that most of them were related to solar activities in their respective publications. The authors then introduced other studies linked such cyclicity to atmospheric circulation such as SAM and ENSO, and rapid climatatic events globally. This part is confusing as it seems related to so many processes and how exactly they work needs better elaboration.

>We agree with both reviewer 1 and 2 that this was unclear in the original paragraph. Therefore this has now been rewritten (lines 351 to 374) and we put forward two possible explanations: 1) SST/SAM variability resulting from internal variability and ocean-atmosphere interactions, 2) solar forcing that influenced both the ocean and atmosphere. We have added more explanation of the link between ocean convection, SST and atmospheric circulation as proposed in Latif et al. (2013). We have added additional references to show that there is evidence for the link between solar forcing and the SAM in the modern period. For clarity and to avoid too many possible causes being suggested we have removed the part about ENSO and rapid climate events globally, as these were perhaps the most speculative parts of this paragraph and were not discussed elsewhere in the paper.

Some minor points are as follows:

>Thank you for these corrections particularly the suggested references

Line 34: For example during ... -> For example, during... Younger Dryas (13.02-11.76 ka BP) - 12.9-11.7 ka (Rasmussen et al., 2014 Quat. Sci. Rev. 106, 14-28)

>This has been corrected

Line 35: accumulate in the South Atlantic -> accumulate in the Southern Ocean causing a "bipolar seesaw" characterized by... -> causing a "bipolar seesaw" pattern characterized by..

>Corrected

Line 63: during the Holocene there is... -> during the Holocene, there is...

>Corrected

Line 71-72: extending from the Subantarctic Front in the north... -> extending from the Polar Front in the north... (Diekmann, 2007, Deep-Sea Res. II 54, 2350-2366)

>Corrected

Line 74: (Park et al., 1998) -> Park et al. (1998) is not a sea ice study. This can be replaced by more recent satellite observations of sea ice extent, such as Parkinson & Cavalieri, 2012, The Cryosphere 6, 871-880.

>Corrected

Line 101-102: ...in the southwestern Indian sector of the Southern Ocean... -> To make the text more concise, this can be removed as there is an extensive description of the study region in the previous chapter.

>Corrected

Line 108: Neogloboquadrina pachyderma -> species name should be italic.

>Corrected

Line 134: Winter sea ice concentration was... -> abbreviation (WSIC) should be noted here

>Corrected

Line 173: A number of references describing environmental preferences of the species need to be mentioned before referring certain species to, e.g., PFZ species, POOZ species, and sea ice species. References can be cited are Zielinski and Gersonde, 1997, Palaeo 3; Crosta et al., 2005, Palaeo 3; Armand et al., 2005, Palaeo 3; Romero et al., 2005, Palaeo 3; Esper et al., 2010, Palaeo 3.

>References have been added (in paragraph from line 182)

Line 234: This record was constrained... -> The MD07-3088 record...

> Corrected

Line248: warming in the COR1GC and ice core records... -> warming in the COR1GC and Antarctic ice core records...

>Corrected

Line 262: Hogg et al., 2007 -> Hogg et al., 2008

>Corrected

Line 299: Reconstructions support that the SWW were weaker between 11 and 7 ka BP... - The authors need to mention that the weakening of SWW at this time interval was north of modern Polar Front. A southward shift of the SWW was inferred during this time interval (as suggested by opal accumulation, Anderson et al., 2009; Xiao et al., 2016), which indicate the SWW in regions further south can be stronger.

> This point has been addressed in major correction 3c

Line 303-304: This is uncertain as modern observations instead show that weaker

SWW's cause reduced eddy activity and less poleward heat transport across the ACC, resulting in cooling (Hogg et al., 2007; Screen et al., 2009). -> Reference error: Hogg et al., 2008 As above, SWW did not weaken at all latitudinal bands.

> The point about SWW weakening at all latitudes has been addressed in correction 3c above, and the reference error has been corrected.

Line 306-309: The duration and timing of the SST cooling observed in the COR1GC record coincides with an AMOC reduction and North Atlantic cooling associated with the 8.2 event (Ellison et al., 2006) however it has not generally been observed in records from the southern hemisphere (Alley and Ágústsdóttir, 2005). - Modeling study by Renssen et al. (2010) suggests the upwelling of cooler NADW in the Southern Ocean could result in the drop of surface temperature between 9 and 7 ka.

>A sentence about this study has now been added to the paragraph (line 330)

Line 315: From the mid to late Holocene the high latitude insolation decreased (Divine et al., 2010) -> Unclear expression. Do you mean southern high latitude? Summer insolation increased during the mid-late Holocene in southern high latitude. Do you mean winter or spring insolation? or annual insolation? Divine et al., 2010 is not the proper reference for insolation, use Laskar et al., 2004 instead.

> It is now clarified that we meant spring southern high latitude insolation, again suggesting that this caused a change in the length of the summer season (line 340). The Laskar et al 2004 citation has also been added

Line 315-316: causing the SWW's to shift northward - coherent with the northward shift of the SWW

> Corrected

Line 317: Divine et al., 2010 -> use Laskar et al., 2004 for insolation reference

>Corrected

Line 317: causing the SWW's to strengthen (e.g. Saunders et al., 2018) -> as above, need to mention strengthening in areas north of the modern Polar Front. It can be weakening in the south.

>We have added a sentence at the end of this paragraph to acknowledge that and also included it in our explanation for why productivity and upwelling did not increase here (lines 348-350).

Line 319-320: The strengthened winds may also have increased upwelling and therefore productivity. -south of the Polar Front, upwelling was reduced during the mid-late Holocene (Anderson et al., 2009).

>This citation and point has been addressed, along with the previous correction, at lines 348-350.

Line 336: in the atmosphere may... -> in the atmosphere circulation

>This line was removed when addressing correction 3e

Line 356: spring insolation extending the summer season -> what does it mean?

>By this we meant that if the spring was warmer then the length of time when the ocean would warm up during the spring and summer period would be extended and the duration of winter cooling reduced. This has previously been suggested by Shevenell et al., 2011 and Etourneau et al., 2013. This is now more fully explained in the discussion at line 315-318. We have reworded the sentence in the conclusion at line 390 to try and make this clearer.

Fig. 2: as the last age control point is at 240.9 cm (13.93 ka BP), why there is a sudden increase in sedimentation at the core base, where no actual age constraint is available?

>This was a mistake and has been corrected.

Fig. 3: I would suggest to arrange the species from left to right according to their ecological preferences in terms of temperature. For example, F. separanda, F. rhombica and T. gracilis are cold water species, which can be placed next to the sea ice species. Such arrangement will show evolution pattern of the diatom assemblage more clearly, which reflect environment changes.

> We feel that the graph as it is best shows the temperature preferences of the species, which are already ordered from left to right in order of water mass, going from cooler POOZ > PF> SA species. We agree however that the sea ice species should be closer to the cold water species for easy comparison, so have moved this to the left of the graph.

Fig. 8: the opal flux record in panel (d) seems incomplete? There are breaks in the curve.

>This has been corrected.

Please also note the supplement to this comment:
https://www.clim-past-discuss.net/cp-2020-23/cp-2020-23-AC1-supplement.pdf

---

## Author Comment (AC2) · 1 May 2020

[note- author responses shown by '>']

Reviewer 2 The sequence of warmings and coolings associated with the last deglaciation and the Holocene has shown contrasting patterns between southern and northern high latitudes. Global-scale processes such as variations of AMOC strength and the alteration of atmospheric circulation seem responsible for this contrast. Most of the high resolution paleorecords studied so far were gained from Antarctic ice cores. Therefore, acquiring high-resolution proxy records of past sea surface temperature is relevant for finding out how spatio-temporal patterns of water temperature evolved

in the Southern Ocean and whether any links with Antarctic temperature variability are recognized. Orme and co-authors present a high-resolution, diatom-based record of sea-surface temperature gained in the western Indian sector of the Southern Ocean (Core KH-107 COR1GC, ca. 54.27◦S, 39.77◦E, WD 2834 m). Their sediment record spans the past 14.2 Ka BP. The age model bases on fifteen AMS radiocarbon dates obtained on mono-specific samples of the planktic foraminifera Neogloboquadrina pachyderma sin. The average temporal resolution of their diatom counts is ca. 60 years (diatom analysis was conducted every cm in the 2.48 m gravity core). By using the Modern Analogue Technique applied to diatom assemblages, Orme and colleagues estimate the summer SST and the winter sea ice concentration. They describe and discuss patterns, timing and magnitude of sea-surface temperature variability through the late deglaciation and the Holocene in the western Indian sector of the Southern Ocean and possible links to global and regional forcings and mechanisms. The Introduction presents basic information on (1) deglacial events and Holocene intervals, and (2) main mechanisms/forcings behind temperature in the Southern Ocean; it reads well and helps the reader less familiar with issues addressed later in the MS. The Methodology is clearly written. Results are concisely presented; the results representation however can improve (see suggestions below). Figures are self-explanatory and necessary in number. References are satisfactory. A major concern is how the Discussion is organized. Throughout the Discussion, there are several inconsistencies and several vague statements that lack scientific support. Some ideas are shortly presented, without any further and deeper discussion. Too many forcings and mechanisms are offered as possible explanations for the SST variations (solar forcing, internal climate variability, sea-ice and productivity changes, ocean-atmosphere coupling, rapid climate change events globally, establishment of modem ENSO amplitude and frequency), without clearly distinguishing which mechanism/forcing/s was/were more important when, and the reader gets lost. Please consider: (1) shortening and focusing the discussion, and (2) adding a Table summarizing with main mechanism/forcing/s for each for each of the discussed intervals.

> We would like to thank reviewer 2 for the very helpful comments provided, which we feel have improved the manuscript.

> In the discussion, particularly the first two paragraphs of section 5.2.1., the text has been shortened to make the discussion more focused. See lines 270 to 290. In line with a similar comment from reviewer 1 (number 3e) and the comment here labelled L326-340 the paragraph about cycles/variability at the end of the discussion (lines 351-374) has been reorganized and re-written. We start by presenting the evidence for 200-300 year cycles in both Southern Ocean records and those reflecting westerly winds/SAM and then use this as a basis for stating two hypotheses 1) that there was shared internal variability and ocean-atmosphere coupling or 2) a shared external forcing (solar). In support for the first point we describe the model findings of Latif et al (2013) about centennial internal variability in the Southern Ocean and atmospheric circulation triggered by changes in deepwater formation. We have added more detail about this. In support of the second point, that solar forcing may have caused the changes, we have linked to modern evidence for the effect of solar activity on the SAM during the instrumental period. We have removed the parts of the discussion about ENSO and rapid climate events which were more speculative, and instead focused on the two causes we consider most likely. > As suggested we have now added a table to summarise the main causes suggested for the Younger Dryas, millennial changes and mid-late Holocene variability (lines 730, table 2) and referred to this in the text (line 268)

Below I list several minor comments and give some suggestions which might be helpful to improve your MSL. 66-67: this is repeated several times throughout the Intro. Please revise

> This sentence has been shortened and merged with the previous sentence to reduce the repetition (lines 66-68).

L. 93-94: since the authors state that 'Topography has a strong influence on the position and form of the ACC in this region', they should provide a more detailed figure

of the study area in Fig. 1, including bathymetric information.

> We have now included in figure 1 a bathymetric map of the Conrad Rise adapted from Ansorge et al (2008) including the current strength (shown by absolute geostrophic velocity).

L: 108: Neogloboquadrina pachyderma should be in italics. > Corrected.

L. 115-125: Using reservoir age from a core gained in a mid-latitude coastal upwelling system is -at the least-risky and caution is advised. Oceanographical and nutrient conditions in the SE Pacific Ocean are quite different from those in the Conrad Rise and can hardly be straightforwardly applied. The high-resolution sampling (every cm) make the age model uncertainties even larger. Additionally, reworking should be considered/discussed.

>The paragraph as a whole has been rephrased to explain the different evidence for reservoir changes in different locations (lines 105-119), which justifies our decision for choosing to use a consistent reservoir age through the record. We hope that this balanced assessment of the limited available evidence shows the reader that there are age uncertainties associated with the selection of reservoir ages. We have included the evidence of reservoir ages from the Siani et al (2013) study mentioned here in this paragraph for couple of reasons. The first is that this site carries a Southern Ocean signal, as it is bathed in waters from the Southern Ocean that are deflected northward by the South American continent, and is outside of the Peruvian upwelling system, with little evidence for local upwelling (e.g. low primary productivity; Abrantes et al., 2007). The second reason is that this is the only core to have reservoir age estimates for both the last deglacial and the Holocene. Therefore while we did not base our chosen reservoir ages on this study, we feel that the consideration of different evidence from the Southern Ocean does help show the reader the current limited and challenging nature of reservoir age estimates. Though there might be some discrepancies in reservoir age changes through time between basins, the good correspondence between COR1GC

SST and ice core records (figure 7) suggests that the corrections made in the present study are adequate.

L. 145-46: these three processes strongly impact your diatom signal (the basis of your SST reconstructions), but it is hardly discussed in 5. Discussion.

>This answer also covers the correction for line 241-243 below. Many studies have shown that diatom sedimentary signals, though imperfect, preserve the main features of diatom productivity/assemblages in surface water (Zielinski and Gersonde, 1998; Gersonde and Zielinski, 2000; Armand et al., 2005; Crosta et al., 2005; Romero et al., 2005). As such, diatoms have been robustly used to quantitatively infer past surface conditions through several statistical techniques (Gersonde and Zielinski, 2000; Crosta et al., 2004, 2020; Nielsen et al., 2004; Esper et al., 2014; Esper and Gersonde, 2014; Ferry et al., 2015; Benz et al., 2016; Xiao et al., 2016), generally in agreement with other techniques based on other micro-fossils or geochemical proxies (Becquey et al., 2002,2003; Panhke et al., 2005; Ho et al., 2016).

> We have added a paragraph to the methods section 3.3 about the dissolution of diatoms (lines 135-146). We feel that this fits better at this point in the paper rather than in the discussion. We observe that the dissolution of poorly silicified diatoms, often those from cold waters, could result in reconstructed warmer temperatures and less sea ice (Xiao et al 2016). However we note also that temperature is the dominant factor effecting species assemblages, and that diatom dissolution is at a minima at 50-55°S (Pichon et al., 1992; Esper et al., 2010). This means that while dissolution may have altered the assemblages to some degree, the effect on the reconstructed temperature should not be strong particularly at the latitude of the Conrad Rise. During analysis we observed good preservation through the core, with (1) well preserved diatom valves in which the fine ornamentation was still visible, (2) diverse diatom assemblages containing diatoms along the whole size spectrum and (3) little fragmentation. Finally, the good general agreement of the new SST record with previous records from similar realms and with the ice core temperature records, gives confidence in that the assemblages in

COR1GC are reflecting climate and that our diatom-based reconstruction are therefore robust.

L. 172: please give age ranges for the Holocene (which age definition of Holocene did you follow?) I suggest adding a box in the upper part of Fig 4, indicating the main intervals of the last deglaciation (YD, ACR, etc.) and Holocene (early/middle/late). This is presented later in Fig 7, but it should be earlier when Results are described.

> We have included reference to the formal defined boundaries of the early Holocene of 11.7-8.2 ka BP (Walker et al., 2018) in the introduction at line 51. We do not separate between the mid and late Holocene in the discussion and results, therefore the age boundaries of these have not been specified or separated in the figures. We have added the lines and labels to figure 4 as suggested, and also added the time periods to figure 3.

L. 173-183: References for the paleoecological information of the diatom species should be provided here. The reader does not know where the species ecology does come from

>This has been corrected (line 182-195)

L. 184: 'The estimated total diatom abundance shows a decreasing trend through the record', please revise this statement. It is not quite correct to state that the total diatom abundance (TDA) shows an overall decreasing between last deglaciation and the latest Holocene. Indeed, TDA varies strongly up to 5.5 Ka BP and experienced afterward a two-step decrease, first around 5.5 Ka BP and later between 4 and 1 ka BP.

> We decided to change this to the diatom flux rather than the diatom abundance. Diatom flux is a closer proxy for productivity and upwelling than diatom abundance because it accounts for changes in sedimentation rate and sediment density. We used the approach of Romero et al. (2015). We had the required data on diatom abundance and sedimentation rate for COR1GC however not the density data for this core. We

therefore used the density data for COR1bPC, which was taken from the same location during the same cruise. As both cores have good chronological constraints (COR1bPC has 13 radiocarbon dates for the last 14.2 ka BP) it was possible to transfer the density values from COR1bPC to COR1GC by identifying the depths that were closest in age. The largest age difference between samples that were transferred was 20-30 years. The results now show a decrease in diatom fluxes at 14.2-12 ka BP, low fluxes from 12-10.3 ka BP, higher fluxes between 10.3 and 5.2 ka BP and lower fluxes after 5.2 ka BP. We have altered the methods (lines 147-154), results (lines 195-198) and discussion at lines 294 and 347 (see also our response to corrections below).

L. 190-198: all short intervals mentioned here should be easily recognizable in Fig 4. Please add some arrows to help the reader to better understand what you are trying to communicate. Moreover, add marks between millennial ages in Fig 4: your Results description goes into centennial-scale description (e.g., Between 11.6 and 8.7 ka BBT, etc). Without these centennial-scale marks is even more difficult to recognize whichever trends and shifts occurred.

>We have added tick points every 200 years to Figure 4 so the reader can assess the centennial timings as suggested. We have added arrows to highlight the two significant centennial events at c.8.2 and c.2.2 ka BP as identified by SiZer and explained in this paragraph. We have also added a line to show the increasing trend during the Younger Dryas. These changes are in addition to the alterations made for the comment below labelled Figure 4.

L. 210: 'high temperatures between 11.6 and 8.7 ka BP during the Early Holocene, followed by a cooling trend thereafter': this is a matter of interpretation. The range of SST variability is larger (larger amplitude) between 8 and 1 ka BP than earlier between 8.2 and 11.8 ka BP. However, is it correct to state that a cooling occurred during the middle to late Holocene? I am not able to recognize a clear decreasing trend in your data depicted in Fig. 7d.

> This sentence is referring to the findings in other records rather than the COR1GC record, although we are stating that these other records show similar findings to the COR1GC record. There is evidence for a slight cooling trend in that the mean decreases from 4.3-3.9°C between the early to mid-late Holocene, there were lower minimum temperatures and reconstructed sea ice increased in the mid-late Holocene but was absent in the early Holocene. However this cooling is shown by the SiZer analysis to not be statistically significant. We have added to the results the word 'slight' in places to show the cooling wasn't large (between lines 202-205) and added a sentence to state that the cooling is not significant in the SiZer analysis section of the results (line 208). In the discussion section 5.1 we have added a sentence to acknowledge that unlike some other records the COR1GC cooling in the Holocene was slight (lines 230) but acknowledge here that there were cool events (low SST excursions, sea ice species). We have also adjusted the wording in the discussion (line 348) and conclusion (lines 379) to acknowledge that the reconstructed temperature difference is minimal.

L. 215: 'Although most records, including COR1GC, show a long-term cooling over the Holocene (Xiao et al., 2016)'. Please revise: the PS2606-6 SST record shows similar values during YD and the entire Holocene. SO, where is the Holocene cooling?

> This sentence is about the Holocene, not the temperature difference between the YD and late Holocene. During the Holocene the PS2606-6 record is showing a cooling, whereby temperatures at c.12-9 ka BP in this record were warmer than the period after 9 ka BP (Xiao et al., 2016). This is in support of other records which generally also show cooling (e.g. Bianchi and Gersonde, 2004; Anderson et al., 2009). This PS2606-6 record shows an abrupt change at 9 ka BP, rather than a gradual cooling, therefore to acknowledge that there is a difference between the gradual cooling shown in some records and the rapid cooling in others, we have added: 'Furthermore, while most records show cooling over the Holocene, either gradually or as an abrupt cooling at the end of the early Holocene (Xiao et al., 2016), there are some differences between records' at line 228.

L. 218: 'The records from Bouvet Islands', where is this? Which sector of the SO? Please provide more accurate information.

>We have added 'in the eastern Atlantic sector of the Southern Ocean' after this statement at line 237.

L. 222-23: 'Our new record from core COR1GC conversely shows SSTs were 1◦C lower during the ACR compared to the mid-late Holocene', Is this 1◦C difference statistically significant? 1◦C of SST difference lays surely within the range of variability of your SST reconstruction. >Given that the temperature difference between the ACR and late Holocene is close to being significant (given the RMSEP of 1°C) and there is a good similarity between the magnitude and patterns of temperature change between the COR1GC SST and ice core records, we feel confident that the cooler temperatures during the ACR are real. However we have added a sentence to acknowledge that transfer function prediction error is a possible cause of the differences between records (such as other records showing no difference in temperature between the ACR and mid-late Holocene), as this is a factor potentially effecting other transfer function based records as well. Lines 243-245: 'The contrasting findings between SST reconstructions may be explained by the reconstructed temperatures being close to the prediction error of SST transfer functions, which is ~1°C in this study and 0.86°C in Xiao et al. (2016).'

L. 233: can a 2-3◦C rise of SST during the Holocene -compared to last deglaciation- as WARM conditions? I understand that it was warmer, but it is not a warm environment per se, mainly when your SST reconstructions is compared with records from mid and low latitudes.

>We did not mean to imply that conditions were warm, rather that they were warmer compared to the rest of the record. We have changed the wording from 'relatively warm conditions through the Holocene' to 'leading to slightly warmer conditions through the Holocene' (line 257 )and also later in the section referred to the 'marginally warmer

early Holocene' (line 260). In the conclusions we have also adjusted the sentence at line 378 to clarify that these temperature changes were slight.

L. 241-43: I agree with these mechanisms and forcings impacting the reconstructed SST record at your core site. However, since diatoms experience dissolution between sea surface and the bottom of the ocean, I can assume that your reconstructed SST values vary depending on which species did it to the sediment. There is, however, no discussion on the possible role of preferential dissolution/preservation of diatoms (see also l. 145-46).

> Please see our response for point L145-146 which addresses this. The discussion of this has been incorporated into the methods section rather than here (line 135-146). We consider that the oceanographic conditions at the site have not changed enough through time to alter the amount of diatom dissolution and therefore the SST signal.

L. 245: 'Southern high latitude warming (Termination 1b) during the Younger Dryas', this is given as one unique interval before (see l. 209-10). Please revise and rephrase correspondingly.

>It is not clear what is meant here, but for clarity we have removed the term 'termination 1b' and instead used the Younger Dryas throughout the paper.

L.266-67: 'Greater upwelling has been shown by higher opal deposition to the south of the Polar Front in the Southern Ocean (Atlantic, Indian and Pacific sectors) through the period 12.7- 11.5 ka BP', this is true. However, your TDA data do not show any sig-nificant difference among ACR, YD, and early Holocene. Therefore, your data offer no convincing evidence of an intensification of upwelling following the last deglaciation.

> This paragraph (now lines 277-290) is about the explaining the identified sequence of events and evidence for this based on previous studies, rather than linking with our evidence which comes in the following paragraph. Therefore we have not changed this

statement but have adjusted the part where we discuss our results (see next comment).

L.275-76: 'as tentatively inferred from the slightly increased diatom abundances at 12.7-12 ka BP'. This is hardly recognizable in your COR1GC record. Your TDA does not actually differ from earlier and later values. Please revise.

>We have changed the diatom abundance to diatom fluxes, as explained above in response to the correction L184. The new diatom flux record however also doesn't support that there was higher diatom productivity or upwelling at this time, as the values decrease from 14.2 to 12 ka BP. Therefore we have altered this sentence to read: 'However there is no evidence of increased productivity and therefore upwelling at the Conrad Rise, as diatom fluxes instead decreased through this period (Figure 8E)' (lines 294-295)

L. 283-84: 'Indeed a southward shift is indicated by an increase in Polar Front species at c. 12 ka BP', you mean Thalassionema nitzschioides var. nitzschioides? The increase is not that clear in F. kerguelensis.

> Yes this was in reference to the increase in Thalassionema nitzschioides var. lanceolata. We have specified this now at line 302

L. 296: 'has been attributed to high annual, winter and spring insolation levels', please clarify: do you mean average annual insolation or winter and spring insolation?

>We have changed this to spring insolation and now cite the papers Shevenell et al. (2011) and Etourneau et al (2013) who also concluded that spring insolation caused early Holocene warming due to a longer summer season. Lines 315-318.

L. 315: SSW's, misspelling. See also below l. 317.

>This has been corrected

L. 316: 'together can explain the gradual cooling in the COR1GC record', please provide SST range, average and 1 STD for your mid and late Holocene SST record.

[Figure]

> We have now ensured that the mean, standard deviation and range is now provided for the three key periods (ACR, early Holocene and mid-late Holocene) in the results section, lines ~200. These show that the mean decreased from 4.3 to 3.9 °C and there were more frequent low temperature excursions in the mid-late Holocene, with the minima changing from 3.3 to 2.2°. We have adjusted the wording as follows at line 342: 'which together can explain the slight cooling in the COR1GC record, the increased frequency of cold events and increase in sea ice'. Which we feel provides the reader with clarity about the evidence for the mid-late Holocene cooling in this record.

L. 320: 'however it is not clear if this occurred as although there was a gradual increase in sedimentation rate, potentially reflecting an increased deposition of diatoms'. Be cautious with this: according to your data, no increase in total diatom abundance occurred at this time.

> We have re-written this sentence to show that the evidence of a decreasing diatom flux does not support increasing productivity, and removed the sentence about the sedimentation rate increasing, which has a less direct link with productivity. At the recommendation of reviewer 1 we have also added a sentence about the reason for this decrease. (lines 346-350)

L. 326-340: this is a quite different story from all the above discussion and confuses the reader. Several forcings are mentioned/shortly discussed in 14 lines (solar forcing, internal climate variability, sea-ice and productivity changes, ocean-atmosphere coupling, rapid climate change events globally, establishment of modem ENSO amplitude and frequency). Presenting an alternative climate scenario at the very end of the manuscript (rapid climate change events globally and establishment of modem ENSO amplitude and frequency), without any further discussion makes this subsection even more confusing and does not add anything valuable to your overall Discussion.

>This point was shared by reviewer 1 and to address it we have re-written this final

paragraph. Please see our response to reviewer 1 and the above comment at the start of this response.

Figures Fig 1: please consider (1) zooming into the closest area to core COR1GC (include bathymetry), and (2) identifying the Atlantic, Indian and Pacific sectors of the Southern Ocean. In the caption the references for cores TN057-13-PC4, TN057-17PC1, MD07-3088, EDML, and EDC should be presented.

>We have now included in figure 1 a bathymetric map of the Conrad Rise adapted from Ansorge et al (2008) including the current strength (shown by absolute geostrophic velocity). We have added labels for the Atlantic, Indian and Pacific sectors to panel (a) and referenced each of these records as recommended.

Fig. 3: note that you name core COR1GC differently depending on the figures. Please revise. var. in Thalassionema nitzschioides var. nitzschioides should not be italics. Thalassiosira oestrupii has been renamed for years already, the current name is Shionodiscus oestrupii. Consider using exponential nomenclature for x-axis of TDA. Consider adding some arrows to lead the reader in better understanding major shifts/changes in (1) the species composition of the diatom assemblage, and (2) total diatom abundance.

>We have changed all the figure captions so they are COR1GC rather than KH-10-7 COR1GC. We have changed Thalassiosira oestrupii to Shionodiscus oestrupii, both here and in the manuscript. We have corrected the labelling so var. is not in italics. We have added arrows to show the major trends in the diatom species abundances and the diatom flux data. The axis label is now in exponential nomenclature.

Fig. 4: Please consider adding a box in the upper panel indicating YD, early/mid/late Holocene, etc. (see Fig 7) Consider also adding the (1) average and 1STD of your own data, and (2) present-day summer and winter SSTs., and (3) present-day mean winter sea ice concentration.

>We have added and separated the sections in the COR1GC record as advised and highlighted the modern summer SST and sea ice concentration. As diatom records reflect summer SST rather than winter SST, we have not included the winter SST as this would have no relevance to the record presented. We also decided not to present the average and standard deviation on the plot, as it was our view that this could make the graph look crowded. The average and standard deviation for the different sections are included in section 4 as advised previously.

Fig. 7: the long-term pattern of your SST data is pretty similar to that of East Antarctica cores: low ACR values, increase during the YD, and warmer Holocene SST. A simple statistical analysis should help you to better understand the trends. Your SST record shows ten SST minima (cooling) during the Holocene: it seems to me that most of these minima are made by only ONE sample. This can be part of regular variability of the diatom assemblage and not at all related with actual SST variations. Caution is advised in the interpretation of these minima!

>The SiZer analysis already represents a step forward to statistically understand the significance of the short-term variability. We feel that additional statistical tests, like providing a correlation matrix, is hampered by the age uncertainty and low resolution of many records. In this vein, a direct correlation test between COR1GC SST record and ice core records, though possible, will imply important resampling and standardization steps that may alter the results and may not bring many new information. However to show more clearly the similar timing of changes between the COR1GC SST record and those from ice cores, we have added a supplementary information file including the COR1GC SST record and EDML temperature ($\delta$18O) and sea ice extent (ssNa) records. These have been normalised and smoothed using SiZer (using the local linear kernel estimator) and a bandwidth of 400 years. The results highlight the close association between the SST, atmospheric temperatures and sea ice extent in the region. The Supplementary Information has been referred to in the paper at lines 251 and 293. Although some of the minima are single data points

they represent temperature excursions of 1-2°C which is above the prediction error for the record. Despite the same methods, material and location, the early Holocene does not have these large oscillations in temperature, supporting that the occurrence of these minima in the mid-late Holocene reflect real climate changes rather than just diatom assemblage variability, which would be present through the whole period. We discuss the most persistent cold excursion at 8.2 ka BP but not any of the later minima specifically, other than in relation to the increasing variability, so feel that these are not over interpreted in the manuscript. As such we have not altered the text.

Please also note the supplement to this comment:
https://www.clim-past-discuss.net/cp-2020-23/cp-2020-23-AC2-supplement.pdf

---

## Author Response (AR2)

**Editor Decision: Publish subject to minor revisions (review by editor) (18 Jun 2020) by Erin McClymont**
**Comments to the Author:**
**Thank you for incorporating the suggestions made by the two reviewers. These included some minor formatting concerns but also some concern about the way in which some of the arguments were being presented (age model, discussion of forcings). Given the complexity of the data and the likely forcings this was not an easy task. However, in the revised manuscript you have been able to incorporate these comments and the discussion is much more clear.**

We would like to thank the editor for this feedback and the minor revisions.

**In Table 2 (the new one) could you clarify for the line on Holocene millennial changes which of the options you provide are for 'warm' and which for 'cool' events? For the mid-late Holocene variability line can you clarify if here you are talking about the ~200 year events (to differentiate from the millennial changes line)?**

We have adjusted the description so that instead of describing the causes during the warm and cool events together (e.g. Length of the summer season: Extended/reduced) we have separated the suggested causes for the slightly warmer Early Holocene and the slightly cooler Mid-late Holocene into two small bullet pointed paragraphs:

> Early Holocene (slightly warmer)
>
> > - Extended summer season
> >
> > - SWW weaker and meridional: increased atmospheric heat transport southwards and reduced Ekman transport of cold water northwards
>
> Mid-late Holocene (slightly cooler)
>
> > - Shorter summer season
> >
> > - SWW stronger and zonal: reduced atmospheric heat transport southwards and increased Ekman transport of cold water northwards

We feel that this clarifies which options apply for warm and cool millennial changes.

We have also specified that the mid-late Holocene variability does include the mechanisms suggested for the 200-260 year cycle, by adjusting the definition in column 1 to read 'Mid-late Holocene centennial variability (inc. the 200-260 year cycle)'

**Line 951: you describe 'a sharp cooling event' at 8.2 ka BP but your preceding text is more tentative about this (this is actually the only time that you use this phrase). Perhaps a more accurate description is that there is a significant cooling event at 8.2 ka BP after which there is greater SST variability? You have confirmed with SiZer that this 8.2 ka cooling is significant; a 'sharp cooling' isn't so obvious because it looks more like the 'slightly cooler' conditions after 8 ka BP are driven by the onset of more frequent cooler events rather than a wholescale shift in the average SST? Could you check your phrasing in the conclusion and ensure that it aligns with your earlier description and also the pattern that you want to highlight?**

We have removed the statement that this was a 'sharp cooling' which was misleading and not in line with our discussion. Instead we have altered the text as suggested to state:

'before a significant, centennial-scale cooling event occurred at c.8.2 ka BP, followed by greater SST variability  through the period 8-1 ka BP.'